# Evolution of aerosol optical depth over China in 2010-2024: increasing importance of meteorological influences

Cheng Fan[1], Gerrit de Leeuw[1,2*], Xiaoxi Yan[1*], Jiantao Dong[3] , Hanqing Kang[4], Chengwei Fang[5], Zhengqiang Li[1], Ying Zhang[1]

1. State Key Laboratory of Remote Sensing and Digital Earth & Key Laboratory of Satellite Remote Sensing of Ministry of Ecology and Environment, Aerospace Information Research Institute, Chinese Academy of Sciences, Beijing 100101, China (Cheng Fan email: fancheng@aircas.ac.cn; ORCID: 0000-0003-1547-9758), (Zhengqiang Li email: lizq@radi.ac.cn; ORCID 0000-0002-7795-3630), (Ying Zhang email: zhang_ying@aircas.ac.cn; ORCID: 0000-0001-5856-1052)
2. Royal Netherlands Meteorological Institute (KNMI), R&D Satellite Observations, P.O.Box 201, 3730AE  De Bilt, The Netherlands (email: g.d.leeuw@hotmail.com & gerrit.de.leeuw@knmi.nl; ORCID: 0000-0002-1649-6333)
3. Satellite Application Center for Ecology and Environment, Ministry of Ecology and Environment of People's Republic of China, Beijing 100094, China (Jiantao Dong email: dong.jt@outlook.com; ORCID: 0000-0002-9877-3743),
4. Key Laboratory for Aerosol-Cloud-Precipitation of China Meteorological Administration, Nanjing University of Information Science and Technology, Nanjing 210044, China (email: hanqingkang@aliyun.com; ORCID: 0000-0002-2005-8395),
5. Cloud-Precipitation Physics and Weather Modification Key Laboratory (CPML), CMA Weather Modification Centre, Beijing 100081, China (Chengwei Fang email: fangcw515@163.com; ORCID: 0000-0001-5099-4646)

*Correspondence to*: Xiaoxi Yan (yanxx@aircas.ac.cn); ORCID: 0000-0002-1725-7131; Gerrit de Leeuw (gerrit.de.leeuw@knmi.nl; ORCID: 0000-0002-1649-6333)

**Abstract**

Time series of MODIS/MAIAC C6.1 aerosol optical depth (AOD) and model-simulated AOD over China were used to determine contributions of meteorological and anthropogenic effects on aerosol variations on monthly and interannual scales. The study covered the period January 2010 - September 2024 with the main focus on five representative areas: NCP, YRD, PRD, HNB and SCB. The time series confirm that emission reduction policy has resulted in the effective reduction of the AOD over China. The large increment of the AOD over the YRD during 2018 - 2021 shows that the influence of meteorological effects on the aerosol load increases as AOD declines. During this period, the potential decrease of the AOD over the NCP was effectively cancelled by unfavorable meteorological effects. Meteorological effects, and their variations, are different over each region. Over the NCP, meteorological effects were mostly unfavorable while, in contrast, over the PRD, meteorological effects were initially unfavorable but had a strong effect after 2016 when they reinforced anthropogenic effects, resulting in a substantial reduction of the AOD. In addition, a strong AOD minimum is observed in 2022, attributed to favorable anthropogenic effects, over some areas reinforced by favorable meteorological effects. Monthly mean AOD patterns were distinctly different

before and after 2016, suggesting that aerosol properties changed in response to emission reduction
policy. In summary, this study highlights the complex interplay between meteorological and
anthropogenic factors in shaping AOD variations across China and demonstrates the increasing
significance of meteorological conditions in modulating China's AOD.
**Keywords**
Aerosol, Remote Sensing, Physical and Chemical processes, Model Simulations, Emission reduction,
Meteorological Effects, China.
**1 Introduction**
An aerosol is a suspension of particles and/or droplets in a gaseous medium, which for atmospheric
aerosol is the air. Aerosol particles can be directly emitted (e.g., sea spray aerosol, dust, soot) or
formed from precursor gases such as $SO_2$, $NO_2$, $NH_3$ and (biogenic) volatile organic compounds,
(B)VOCs. In the atmosphere, aerosol particles can change in size, chemical composition and shape,
due to a variety of chemical and physical processes. The number of aerosol particles varies strongly in
space and time due to emissions, gas-to-particle conversion, condensation of vapors, coagulation,
cloud processing, removal by wet or dry deposition, etc. (Seinfeld et al., 2016). In addition, the
concentrations of aerosol particles and their size distributions are influenced by meteorological
processes, in particular horizontal and vertical transport by the wind and convective processes in the
boundary layer, turbulent deposition, hygroscopic growth. Effects of changes in meteorological
parameters on AOD were explained in De Leeuw et al. (2023) and this information was used by these
authors to explain changes in AOD time series (their Section 3.6). Regional and long-range transport
from aerosol source regions, such as biomass burning, industrial, urban or desert areas, or stagnant
weather conditions leading to accumulation, are important phenomena influencing aerosol
concentrations which in turn are influenced by local, regional and large scale weather phenomena. As
a result, the aerosol properties vary strongly in both space and time.
Aerosol particles vary in size from less than 1 nm to more than 50 μm and concentrations vary over
many orders of magnitude, depending on size and on their vicinity to source regions. Particle
concentrations can be measured in many ways. For environmental monitoring purposes, particle mass
concentration $PM_{2.5}$ is often used, i.e. the dry mass of aerosol particles measured with an ambient size
smaller than 2.5 μm. These small particles are important, for instance because of their effects on
human health (Van Donkelaar et al., 2015; Bai et al., 2024). Particles with diameters on the order of
the wavelength of solar light are important because of their interaction with solar radiation and thus
climate. Scattering of solar radiation reduces the intensity of incoming light and thus the warming
effect of greenhouse gases, whereas absorption of solar radiation may cause local heating which
affects meteorological processes. Furthermore, aerosol particles may act as cloud condensation nuclei
and thus affect cloud micro- and macro-physical properties. To obtain aerosol information for climate
applications and aerosol-cloud interaction, sensors are used that measure the intensity of scattered
radiation, i.e. optical instruments such as radiometers or spectrometers. Such instruments measure the
aerosol extinction integrated over the whole atmospheric column, i.e., the aerosol optical depth (AOD)
and other aerosol parameters. They can be used at experimental and monitoring sites providing
information valid for a relatively small area around the site. They can also be mounted onboard
satellites and provide information on regional and global scales.
AOD and $PM_{2.5}$ are both measures for the aerosol concentration, but based on different physical
principles and are therefore not directly comparable. $PM_{2.5}$ is the aerosol dry mass concentration
representative for a small area around the measurement location, usually near the surface. AOD is a
column-integrated quantity, representing aerosol in ambient conditions and subject to meteorological
conditions across the whole atmospheric column. Aerosol properties may vary across the column and
may occur in disconnected layers with different origin and thus aerosol properties. Hence AOD and
$PM_{2.5}$ cannot be directly compared and may respond in different ways to changing meteorological
conditions. $PM_{2.5}$ can be derived from AOD by taking these differences into account, such as by using
a comprehensive transport model (e.g., (Ma et al., 2014; Van Donkelaar et al., 2015; Van Donkelaar et
al., 2021; Zhang et al., 2020; Xiao et al., 2021). $PM_{2.5}$ is commonly used in air quality studies as a
measure for the amount of aerosols (Zhang et al., 2019c). AOD is commonly used in climate studies as
a measure for the amount of aerosol in aerosol-radiation-cloud-precipitation interactions (Rosenfeld et
al., 2014; Kinne, 2019; Bellouin et al., 2020; Liu et al., 2024a).
Satellite observations provide information on the spatiotemporal variation of trace gases and aerosols
in the atmosphere on local, regional and global scales with daily global coverage, provided that
retrievals are successful. The availability of successful retrievals depends on sky conditions (clear for
aerosol, low cloud cover for trace gases), surface conditions, solar zenith angles, etc., which in turn
vary with season and latitude. Satellite data have been used to retrieve AOD since 50 years and long
time series are available from individual sensors such as the MODerate resolution Imaging
Spectroradiometer (MODIS) and combinations of sensors (Sogacheva et al., 2020). Likewise, satellite
information on trace gases, important for the formation of aerosol particles from precursor gases, has
been available since several decades. The use of satellites to monitor the evolution of concentrations of
aerosols and precursor trace gases over China has been demonstrated for, among others, $SO_2$ (Zhang et
al., 2019b; Yan and Xu, 2021; Van Der A et al., 2017), $NO_2$ (Van Der A et al., 2017; De Foy et al.,
2016; Fan et al., 2021; Zhang et al., 2024; Liu et al., 2017), AOD (Xu et al., 2015; Kang et al., 2016;
Zhang et al., 2017a; Zhao et al., 2017; Proestakis et al., 2018; De Leeuw et al., 2018; De Leeuw et al.,
2022; De Leeuw et al., 2023; Sogacheva et al., 2018a; Sogacheva et al., 2018b) and AOD-derived
aerosol mass concentrations ($PM_{2.5}$) (Ma et al., 2014; Van Donkelaar et al., 2015; Van Donkelaar et al.,
2021; Zhang et al., 2020; Xiao et al., 2021).

Long time series of aerosols and trace gases provide information on the evolution of their atmospheric concentrations which are influenced by anthropogenic and natural emissions, atmospheric transformations and removal processes. Anthropogenic effects include emissions due to, e.g., industrialization, urbanization, traffic, domestic activities and associated increase in energy production, transportation, agricultural activities, land use, etc. Emissions, and thus concentrations, are reduced by the implementation of policies aimed at the reduction of air pollution and its adverse effects.

Early in the 21st century, aerosol and trace gas concentrations over China were among the highest worldwide as a consequence of economic development and urbanization. To abate air pollution, a number of plans to reduce emissions of aerosols and trace gases has been implemented in China. As a result, aerosol concentrations have been reduced to below those in 2000, as evidenced from time series of satellite-derived AOD (De Leeuw et al., 2023). Both ground-based (Zheng et al., 2017; Zhang et al., 2019c; Xiao et al., 2020; Geng et al., 2024; Zhong et al., 2021; Zhang et al., 2019d) and satellite measurements show that accelerated efforts resulted in an initially fast reduction of aerosol concentrations between 2011 and 2018.

However, as mentioned above, the concentrations of aerosols and trace gases are influenced by both meteorological and anthropogenic effects, including emission reduction policy, as shown by, e.g., Kang et al. (2019). De Leeuw et al. (2023), using methods similar to those of Kang et al. (2019), showed that meteorological effects were responsible for as much as one quarter of the total reduction of the AOD over the Yangtze River Delta (YRD) between July 2011 and February 2020. The total reduction amounted to 31.4%, due to contributions from meteorological and anthropogenic effects, resulting in an AOD smaller than that in 2000. Over the North China Plain (NCP) the total reduction was 27.2% with 6% attributed to meteorological effects, over the Pearl River Delta (PRD), Hunan and Hubei (HNB) and the Sichuan Basin (SCB) the total reduction was 22.2%, 35.9% and 40.3%, respectively, with 22%, 10% and 17% of the total reduction attributed to meteorological effects.

These results clearly confirm the importance of meteorological effects on the variation of aerosol concentrations which need to be taken into account for the evaluation of effects of emission reduction. Meteorological effects can enhance the AOD and thus counteract effects of emission reduction on the concentrations (unfavorable meteorological effects) but they can also reduce AOD and thus reinforce emission reduction effects on the aerosol concentrations (favorable).

The current study extends the work presented in De Leeuw et al. (2023) for the time period 2010-2021 with almost 3 years by adding MAIAC AOD data from the end of 2021 to until September 2024, over all five regions (NCP, YRD, PRD, HNB and SCB), but with some major differences. In the first place, the MODIS/MAIAC C6 AOD data was not extended beyond 2022 and, therefore, in the current study, the C6 time series for 2010-2021 used by De Leeuw et al. (2023) was replaced with the recently released (6 July 2022) MODIS/MAIAC C6.1 data and extended with C6.1 data until September 2024. Comparison between MODIS/MAIAC C6.1 AOD and the MODIS/MAIAC C6 AOD shows small differences across China in both space and time (Huang et al., 2024). Because of these differences,

especially the adjustments around 30°N (Sect. 2.2), the results mentioned above from De Leeuw et al.
(2023) are somewhat different from those produced in the current study (Fan et al., 2025; in
preparation). In the second place, the KZ(12,3) filter (defined as 3 applications of a moving average of
the values in a window with a length of 12 months) was replaced with the centered moving average
over 12 months (CMA12)[1], filtering variations with a period of up to 12 months instead of 21 months.
The CMA12 time series reveals tendencies and variations which were further investigated using
monthly mean data. One motivation to extend the work by De Leeuw et al. (2023) was to investigate
the suggested flattening of the AOD during 2018 - 2021. Thus, AOD time series are presented for the
period January 2010 - September 2024, primarily as monthly averages of satellite observations.
Following Kang et al. (2019) and De Leeuw et al. (2023), meteorological effects on the AOD were
simulated using the Community Earth System Model (CESM) with emissions fixed in 2010 but with
varying meteorological data nudged to MERRA-2 reanalysis data. Comparison of model and satellite
monthly mean AOD time series shows similarities and thus meteorological effects on the observed
AOD. Differences between modelled and observed AOD are attributed to anthropogenic effects. In
addition to these monthly variations, AOD tendencies and contributions from meteorological and
anthropogenic effects are discussed based on normalized[2] CMA12 time series of satellite and model
data. CMA12 effectively removes monthly and seasonal variations, but also smooths the variations
and introduces uncertainty in the times when events occur.
The results presented below clearly confirm that accelerated efforts resulted in an initially fast
reduction of aerosol concentrations between 2011 and 2018, as presented in the literature for both
$PM_{2.5}$ (Zhang et al., 2019c; Liu et al., 2024a) and AOD (De Leeuw et al., 2023; De Leeuw et al., 2018;
Sogacheva et al., 2018a; Sogacheva et al., 2018b; Zhao et al., 2017; Zhang et al., 2017a). The
reduction of the aerosol concentrations is generally attributed to China's emission reduction policy, but
anomalies are observed for both $PM_{2.5}$ (Du et al., 2022) and AOD (De Leeuw et al., 2024) which, at
least in part, can be attributed to meteorological effects. With the reduction of the aerosol
concentrations, the meteorologically-induced anomalies become relatively more important and may be
of similar or larger magnitude as anthropogenic effects, resulting in a net zero change in AOD, as
shown in this paper.
The objectives of the current study are (1) to investigate the reasons for the flattening of the AOD
reduction during 2017-2021, observed by De Leeuw et al. (2023); (2) to investigate what caused the
anomalous AOD in the winter of 2014 over the YRD, HNB and PRD, but not over the NCP and SCB
(De Leeuw et al., 2023)(Fig. 7); (3) to use monthly mean AOD data to accurately identify the start and

---

[1] Centered moving average over 12 months, which for July is: (AVERAGE(Jan:Dec) + AVERAGE(Feb:Jan))/2
[2] Both model and satellite AOD time series were divided by their respective values in July 2010, i.e. at the start
of the normalized time series, each of the normalized time series has the value 1, as illustrated in Figs. 5, 7, 9, 11
and 13. If there are no meteorological effects on the AOD, the model data points in the time series are all 1; any
deviation from 1 indicates meteorological influences on the AOD. Any deviation between the satellite and model
data indicates anthropogenic influences on the AOD.

end of anomalous events, which are hidden in the low-pass filtered data used in De Leeuw et al.
(2023); (4) to connect the occurrences of anomalous AOD to specific meteorological conditions and/or
anthropogenic interferences; (5) to investigate whether changes in aerosol physicochemical
characteristics, in response to emission reduction and climate change, results in different AOD patterns.
Obviously, not all of these questions can be fully addressed in a single study. In the current paper we
report and describe the observational data and provide comparisons with the CESM model data (with
emissions fixed in 2010). Meteorological and anthropogenic effects on the AOD variations are
discussed and possible influences of large scale meteorological effects (El Niño, La Niña, heat waves)
and anthropogenic effects (policy measures and economic effects) are indicated. These effects will be
discussed in more detail in a follow-up paper. In Section 2, we briefly describe the study area, with 5
selected regions, the MODIS/MAIAC C6.1 data set and the CESM model used in this study. Time
series of the MODIS/MAIAC C6.1 monthly mean AOD and the simulated AOD are presented and
discussed in Section 3, together with time series of centered moving averages, normalized in July 2010,
of both the MODIS/MAIAC AOD and the CESM simulated AOD. Meteorological and anthropogenic
effects on AOD temporal variation, derived from the normalized CMA12 time series, are presented for
each region. The results are discussed in Section 4, with a focus on features common to the five
regions. Conclusions are summarized in Section 5.
**2 Methods**
**2.1 Study area**
The coarse resolution of the CESM model (Section 2.3) used in this study requires relatively large
study areas. The study areas selected are the North China Plain (NCP), the Yangtze River Delta (YRD),
the Pearl River Delta (PRD), Hunan and Hubei (HNB) and the Sichuan Basin (SCB). Their locations
are indicated in the map of southeast Asia, overlaid over the spatial distribution of the annual mean
AOD in 2014, with coordinates presented in Table 1. These five study areas were selected because of
their high population density and industrial activity. They are situated in different climate zones with
different geography, different meteorological conditions and influences from large scale circulation,
different aerosol conditions and influences from long-range transport. The AOD spatial distributions
over the different regions show some sharp transitions, such as over the NCP where mountains to the
north and to the west are blocking transport of atmospheric constituents, resulting in a large spatial
gradient. Likewise, the SCB is a basin surrounded by high mountains which prevent ventilation and
thus atmospheric constituents are trapped. The AOD map also shows the large AOD differences
between the five regions, with high AOD over most of the SW of the NCP and a north-south gradient
leading into the highest AOD in the north of Henan. Over the YRD, a large north-south gradient is
observed, but with the highest AOD in the north (Jiangsu and Anhui) and substantially lower AOD in
Zhejiang. In contrast, the AOD in the PRD is mainly centered around the urban area of Guangzhou

which is situated in the central and southern part of Guangdong province, in the north of the Pearl River Delta. The prevailing northerly wind in the winter facilitates the transport of air pollution whereas the southerly wind in the summer brings clean air from the South China Sea (Liu et al., 2020b).

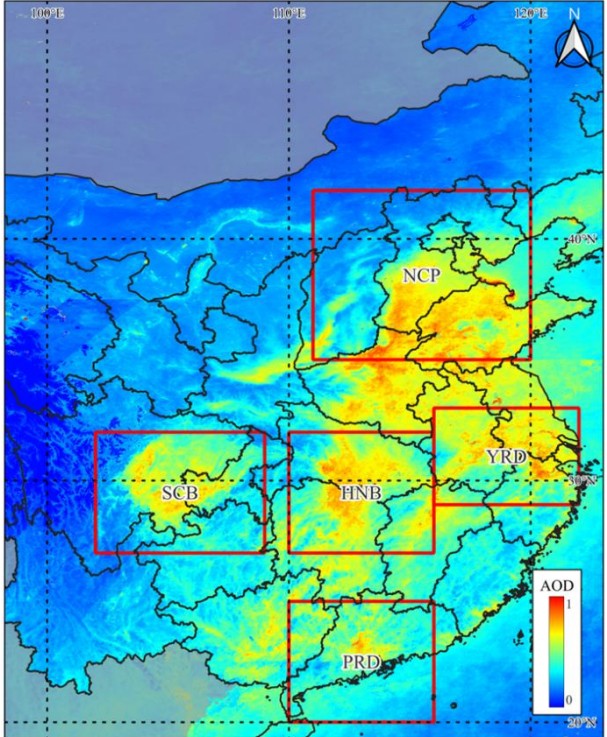

Figure 1: Map showing the study area in southeast Asia, overlaid on the annual mean AOD in 2014 (see legend for color scale). The five selected study areas are indicated by the rectangles, their coordinates are indicated in Table 1.

Table 1. Five study areas were selected in China as shown in Fig. 1. Each area is defined by the latitude and longitude at the lower-left and upper-right corner.

|  | NCP | YRD | PRD | SCB | HNB |
|---|---|---|---|---|---|
| latitude of the lower-left corner | 35°N | 29°N | 20°N | 27°N | 27°N |
| longitude of the lower-left corner | 111°E | 116°E | 110°E | 102°E | 110°E |
| latitude of the upper-right corner | 42°N | 33°N | 25°N | 32°N | 32°N |
| longitude of the upper-right corner | 120°E | 122°E | 116°E | 109°E | 116°E |

## 2.2 MODIS/MAIAC

MODIS spectrometers were launched on board the Terra satellite in December 1999 and on board the Aqua satellite in May 2002. Terra flies in a near-polar sun synchronous orbit in a descending mode with a daytime equator crossing at 10:30 local time (LT) and Aqua flies in an ascending mode with an equator crossing at 13:30 LT. MODIS is a single view instrument with a swath width of 2330 km across track and a nominal pixel resolution at nadir of 250m (2 bands), 500m (5 bands) and 1000m (29 bands). The 36 wavebands cover wavelengths between 405 nm and 14.28 μm. MODIS has been designed for the retrieval of aerosol and cloud properties and several algorithms have been developed for this purpose, of which the dark target (DT) (Levy et al., 2013), deep blue (DB) (Sayer et al., 2014;

Sayer et al., 2015; Sayer et al., 2019; Hsu et al., 2013; Hsu et al., 2019), the merged dark target/deep blue (DTDB) (Sayer et al., 2014) and the Multi-Angle Implementation of Atmospheric Correction (MAIAC) algorithm (Lyapustin et al., 2018) products are most widely used. Aerosol products from these algorithms are publicly available from the Land Processes Distributed Active Archive Center (LPDAAC) website (https://lpdaac.usgs.gov/, last access 24 Feb 2025) within the NASA Earth Observing System Data and Information System (EOSDIS).

In this study, the AOD at 550 nm (in this paper further referred to as AOD) is used, based on the daily L2 MAIAC C6.1 AOD retrieval products downloaded from the LPDAAC website. Monthly mean AOD was calculated using the Google Earth Engine (GEE) cloud computing platform (https://earthengine.google.com/last access 24 Feb 2025).

The MAIAC algorithm (Lyapustin et al., 2018) produces a combined MODIS/Terra and MODIS/Aqua gridded L2 daily AOD product at 1 km resolution on a sinusoidal grid. The MAIAC algorithm retrieves AOD using MODIS L1B data gridded on a fixed 1 km grid accumulated over 16 days with a sliding window technique. The algorithm effectively separates surface and atmospheric contributions to the TOA reflectance by using observations of the same grid at different times and at different angles from different orbits (Lyapustin et al., 2018). The MAIAC MCD19A2.061 (C6.1, further referred to as MAIAC C6.1 or simply MAIAC) product was released on 6 July 2022. MAIAC C6.1 and differences from previous versions are described by Lyapustin and Wang (2022) , LPDAAC (2024) and Huang et al. (2024). In C6.1, the AOD discontinuity around 30°N (De Leeuw et al., 2022), has been addressed by implementing a gradual transition between aerosol models over a buffer area of 300 km. MAIAC C6.1 does not attempt retrieval over snow-covered surfaces and the ice mask is unreliable (Lyapustin and Wang, 2022).

MAIAC C6.1 has been validated over China by Ji et al. (2024) and Huang et al. (2024). Both studies report that the overall accuracy of the MAIAC AOD products over China is good. The validation by Ji et al. (2024) over bright surfaces, using publicly available reference data from AERONET and CARSNET until 2014, shows a significant underestimation and negative bias of the MAIAC C6.1 product, which however performs slightly better than DB and C6. The comparison with collocated AERONET AOD data, for the period from 2001 to 2021, by Huang et al. (2024) shows good consistency, with correlation coefficients (R) of 0.933/0.939, root mean square error (RMSE) of 0.152/0.146, bias of 0.005/0.015, mean absolute error (MAE) of 0.094/0.092, relative mean bias (RMB) of 1.221/1.301 and percentage of data points within expected error (EE) of 71.02/68.36. These statistical metrics refer to comparison at the overpass times of the Aqua (13:30 LT) and Terra (10:30 satellites, respectively (Huang et al., 2024)(Fig. 2). The comparison shows a slight overestimation of C6.1 at low AOD (<0.5) and a small underestimation at higher AOD.

**2.3 CESM Model**

Meteorologically-induced AOD variations were determined by using the Community Earth System Model (CESM) Version 1.0.4 (Hurrell et al., 2013) with the Community Atmospheric Model version 5 (CAM5) (Neale et al., 2012). CESM has a spatial resolution of 1.9° × 2.5° (latitude × longitude) and 56 vertical levels from the surface to 4 hPa. Concentrations of aerosol components, including sulfate, ammonium nitrate, black carbon, primary and secondary organic aerosol, dust and sea salt, are calculated based on the MOZART-4 (Model for Ozone and Related chemical Tracers version 4) chemical mechanism (Emmons et al., 2010). Model performance on aerosol has been widely evaluated (Lamarque et al., 2012; Fang et al., 2020; Emmons et al., 2010).

To isolate the effects of meteorology on AOD, anthropogenic emissions were fixed using the monthly values from 2010, which were repeatedly applied to the corresponding months of each subsequent year. Meteorological input fields, including horizontal winds, air temperature, surface pressure, land surface temperature, heat fluxes, and wind stresses, were nudged to the MERRA-2 (Modern Era Retrospective analysis for Research and Applications, Version 2) reanalysis dataset (Gelaro et al., 2017; Rienecker et al., 2011) (see also https://rda.ucar.edu/datasets/d313003/; last access 4 March 2024), which provides data at a 3-hour temporal resolution. In this study, we used the MERRA-2 product available at 1.9° × 2.5° horizontal resolution, which matches the CESM model grid and avoids the need for spatial interpolation. Linear interpolation in time was applied between input steps to ensure continuity and avoid artificial jumps (Lamarque et al., 2012). CAM5 employs a sub-stepping algorithm (Lauritzen et al., 2011; Lamarque et al., 2012) and an atmospheric mass fixer (Rotman et al., 2004) to maintain consistency between nudged and prognostic fields. The importance of nudging was discussed in, e.g., Menut et al. (2024); Zhang et al. (2014) and He et al. (2015).

Natural emissions of dust and sea salt were calculated online in the model using the actual MERRA-2 meteorological conditions. Biomass burning emissions from the Global Fire Emissions Database version 2 (GFEDv2) (Randerson et al., 2006) were treated as anthropogenic (Yan et al., 2006; Wu et al., 2020) and fixed at the 2010 level. As a result, all variations in the simulated AOD can be attributed to changes in meteorological parameters and their influence on natural aerosol processes. This approach is commonly used with different types of models (Xiao et al., 2021; Qi et al., 2022; Ji et al., 2020; Zhao et al., 2021), including CESM / CAMS (Banks et al., 2022; Kang et al., 2019; De Leeuw et al., 2023). Model resolution was addressed by, e.g., Bacmeister et al. (2014); Huang et al. (2016); Glotfelty et al. (2017).

AOD was calculated from the concentrations of the aerosol species at each grid point and for each time step with the model developed by Zhang et al. (2017b). The spatial distribution of the thus calculated AOD is similar to that of the MAIAC-retrieved AOD. However, the CESM estimates for desert dust are too high (Wu et al., 2019) and therefore contributions from desert dust were not

included in the AOD calculations. Differences between simulated and MAIAC-retrieved monthly mean AOD over each region will be discussed in Section 3.3.

The model simulations were made for the period from January 2009 to July 2023 (14 years and 7 months), with the first year used as spin-up time. The end date of July 2023 was determined by the availability of the reanalysis meteorological data used in the simulation at the time of this study (August 2024).

## 3 Results

### 3.1 Data overview

MAIAC C6.1 monthly mean AOD time series over the 5 regions are presented in Fig. 2. The data in Fig. 2 shows very high AOD peaks in June 2014 over the HNB (1.07) and the YRD (1.05), in June 2012 over the YRD (0.97) and over HNB (0.90) and in June 2010 over the HNB (0.93). Other high peaks with an AOD of 0.8 are observed in 2010 over the SCB in February and over the NCP in June, in 2011 over the HNB (0.80) in February, over the YRD (0.86) in June and over the NCP (0.84) in August and in April 2024 over the PRD (0.80). The AOD time series over each region will be discussed in detail in Section 3.3.

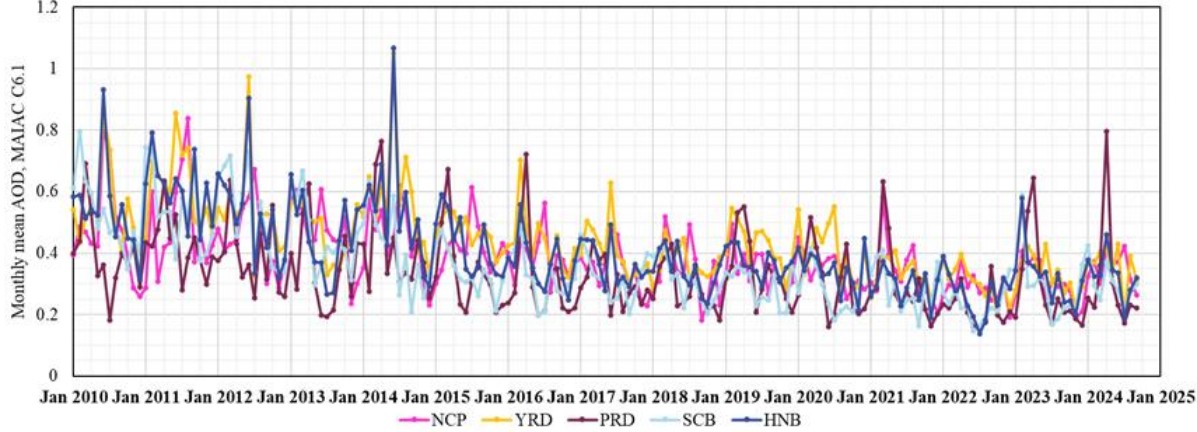

**Figure 2: Time series of monthly mean MAIAC C6.1 AOD data for all five regions (see legend).**

### 3.2 Common features and tendencies

The overall AOD variations between 2010 and 2024 are clearly observed in the centered moving average of the monthly mean observational data, CMA12, which acts as a low-pass filter and removes short-term variations with cycles of up to 12 months, while retaining long term variations. The CMA12 filtered monthly mean AOD data for the five regions in Fig. 3 clearly show the differences between the five regions which are difficult to see in the monthly mean data in Fig. 2. For instance, the high AOD during the first 5 years appears to cover two periods with substantially enhanced AOD, over the YRD, HNB and PRD, separated by a short period with very low values in 2013. In contrast, the AOD over the NCP decreased monotonously between 2012 and 2018, with a small enhancement in

2014 which coincided with the elevated AOD in other regions. Over the SCB, the AOD decreased in
2012 at a rate similar to that over the YRD and the HNB, and in 2014 at a faster rate, with a shoulder
in 2013.
In 2014, very high AOD occurred over both the YRD and the HNB, peaking in April/May (0.60;
referring to CMA12) and March (0.58), respectively, with a gradual decrease toward the end of the
year when the filtered AOD dropped substantially during the next 3 - 4 months. A broad maximum
also occurred over the PRD, with high AOD between February and September, but less extreme (0.42).
Obviously, the CMA12 AOD over the HNB and YRD in 2014 was influenced by the extremely high
monthly mean AOD peaks in June (Fig. 2), but these peaks did not have a large effect on the variation
of the CMA12 AOD in 2014 (Figure 3). Replacement of the monthly mean peak values in June (1.07
and 1.05, respectively), with the local mean value of 0.5 resulted in  lower values of the CMA12 but
did not substantially change the shapes of the CMA12 time series during the 12 months affected. The
data in Fig. 2 show that the AOD was also high during preceding months and the CMA12 data in Fig.
3 show the gradual increase from the minimum in the spring of 2013 to the maximum in 2014. This
has been attributed to anomalous meteorological situations during 2014 (De Leeuw et al., 2024).
A third period (2015 - 2018) shows the accelerated AOD decrease, attributed to the effective
implementation of the 2013 - 2017 Clean Air Action Plan. The data show the differences and
similarities in the tendencies and rates of the AOD reduction in the five regions, as well as fluctuations
which may be due to meteorological influences as discussed in the following sections. It is noted that
over the SCB the largest reduction was achieved during the period 2013 - 2015 (see Section 3.3.5).
A fourth period, between the summer of 2018 and the end of 2020 shows the strong enhancement of
the AOD, peaking in 2019, over the YRD, HNB and PRD, while over the NCP the AOD increase was
smaller and over the SCB there was no increase. This period was followed by a strong reduction over
all 5 regions with a minimum in 2022, and recovery in 2023 when the AOD reached a maximum early
in the year, higher than in 2021 over the HNB and the SCB. Over these regions and also over the PRD,
the AOD decreased in the second half of 2023 and over all five regions the AOD remained high until
March 2024 (the end of the CMA12 of the MAIAC C6.1 AOD time series in this study).

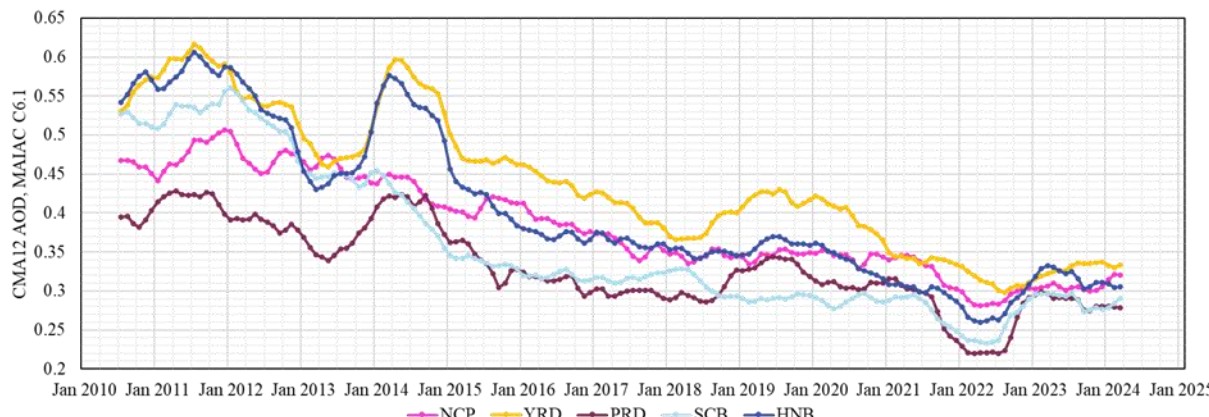


Figure 3. Time series of CMA12 filtered monthly mean AOD for the five regions (see legend) from July 2010 until March 2024.

Detailed comparisons between the monthly mean AOD from MAIAC, the CESM model simulations and the CMA12 filtered data are presented in the following Sections, for each of the 5 regions separately. CMA12 time series are used to visualize tendencies. However, as discussed in Kang et al. (2019), the CESM results are not representative for actual situations because AOD was simulated using fixed emissions to identify meteorological effects on the AOD. The CMA12 filtered observational and simulated AOD data were normalized for quantitative comparison and to determine anthropogenic and meteorological influences, as explained in De Leeuw et al. (2023). The anthropogenic effect ($A_i$) on the AOD has been deduced from the normalized observations ($O_i$) and model simulations ($M_i$) following the method presented in Kang et al. (2019), using:

$$A_i = (O_i - M_i)/ M_i \qquad (1)$$

At the start of the normalized time series, $i = 1$, $A_i = 0$, $O_i = 1$ and $M_i = 1$. Normalized model-simulated AOD larger than 1, i.e. enhanced AOD, results from unfavorable meteorological effects. Unfavorable meteorological effects offset the effects of emission reductions aimed at decreasing aerosol concentrations and thus AOD. Vice-versa, favorable meteorological effects reduce AOD and thus reinforce emission reduction policy. Normalized anthropogenic effects larger than zero results from unfavorable anthropogenic effects (such as increased emissions) and effective emission reduction renders favorable anthropogenic effects ($A_i < 0$).

### 3.3 AOD time series over individual regions

In the following Sections, MAIAC monthly mean AOD time series are presented for January 2010 - September 2024 and simulated monthly mean AOD time series for January 2010 - July 2023, together with CMA12 filtered monthly mean MAIAC AOD time series from July 2010 to March 2024, for each of the five regions separately. In addition, normalized (to July 2010) CMA12-filtered time series of MAIAC and simulated data, showing meteorological contributions to the AOD, are presented for July 2010 to January 2023, together with anthropogenic contributions calculated using eq. (1).

### 3.3.1 AOD time series over the NCP

Time series of the monthly mean MAIAC and CESM AOD over the NCP in Fig. 4 show that, overall, the observed AOD variations are qualitatively reasonably well reproduced by the CESM model, but not in a quantitative sense. A discrepancy was expected because in the model simulations the emissions were fixed to those in 2010. During the first years, the simulated AOD is substantially lower than the observations, except in November/December 2010 and 2012-2015, when simulated and observed AOD are in good agreement. The discrepancies during the spring and summer are attributed to the omission of the effect of desert dust in the AOD calculations (Section 2.3), while also anomalous meteorological conditions may have influenced the aerosol properties in the NCP (Fang et

al., 2020). Desert dust strongly contributes to the AOD over the NCP (Proestakis et al., 2018; De Leeuw et al., 2018). The better agreement between the simulated and observed AOD during the period 2015 - 2020 is attributed to the reduction of the observed aerosol concentrations in response to the 2013 - 2017 Clean Air Action Plan (which initially did not result in a reduction of the AOD, possibly because of unfavorable meteorological effects as discussed below), while the contribution from dust is not influenced by these measures. Further reduction of anthropogenic emissions after 2020 resulted in increased differences between the observations and the CESM model simulations.

The model-simulated AOD over the NCP varies seasonally with maxima mostly occurring in June and minima in December, often followed by a strong increase into January. Deviations from this pattern are observed, such as during 2013 and the first half of 2014 when strong fluctuations occurred with overall higher AOD, followed by a deep minimum in December 2014 (AOD 0.23 as compared with overall minima on the order of 0.26 - 0.31). Also, in 2011 and 2018/2019 the patterns were more variable than in other years.

The MAIAC observations initially followed a similar pattern, with high AOD peaks during the summer (June/July) and deep minima during the winter which may have lasted a few months. Anomalously high peaks in June 2010 and in August 2011 were followed by a steep descent into September and during the next 6 months the AOD remained relatively high (0.42) and variations were small. During 2013 and 2014, the variation of the monthly mean satellite data was qualitatively similar to that of the model simulations, i.e. fluctuations occurred with overall somewhat higher AOD than in previous years, followed by a deep minimum in December 2014 and a maximum of 0.61 in July 2015. Thereafter, the peak values decreased, the minimum values decreased too but were higher than before 2015. The pattern changed dramatically from 2018. In 2018 the monthly mean AOD varied from month to month and thereafter the AOD peaked in winter (January - March) instead of summer. Winter peaks were also observed in 2011, 2013 and 2014, but these were overshadowed by the high maxima in the summer.

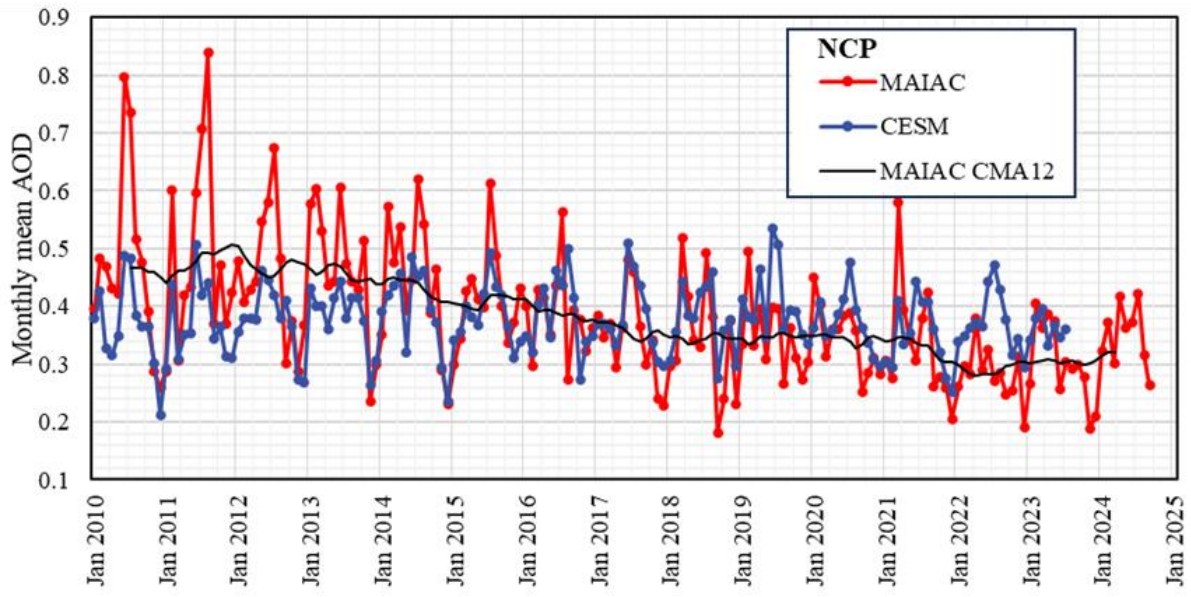

416

**Figure 4: Time series of monthly mean observed (MAIAC) and simulated (CESM) AOD, from January 2010 to September 2024 and July 2023, respectively, over the NCP, together with the MAIAC CMA12 filtered data.**

Fig. 5 shows normalized time series over the NCP of CMA12 filtered MAIAC monthly mean AOD and CESM-simulated monthly mean AOD. The model data show an overall unfavorable meteorological influence (except in 2021), offsetting anthropogenic effects such as due to emission reduction policy. The latter resulted in effective reduction of the normalized AOD between 2010 and 2013, which varied around 1, as the CMA12 filtered MAIAC observations show. These data also show that the AOD decreased monotonously by about 30%, with some fluctuations, from the maximum value at the end of 2012, until a steady value was reached in the summer of 2018. This decrease is attributed to the successful implementation of the 2013 - 2017 Clean Air Action Plan. The different behavior of the model simulations and the satellite data implies that emissions were reduced by anthropogenic effects. However, the AOD reduction was offset due to unfavorable meteorological influences, as indicated by the model simulations. Anthropogenic and meteorological effects were of similar magnitude in 2012 and early 2013 and thus the net effect was close to zero. Fluctuations such as those in 2014, 2015/2016 and 2017 coincide with variations in the simulated data and are attributed to meteorological effects.

Between 2018 and 2021 the AOD changed very little, due to unfavorable meteorological effects. The anthropogenic data in Fig. 5 show that emission reduction was effective until February 2019, but the effect on AOD was very small due to the offset by unfavorable meteorological effects. From February 2019, the unfavorable meteorological effects continued to increase until June, then decreased while the anthropogenic effects became less favorable. As a result, the observed AOD increased somewhat until August and remained relatively constant until May 2021, when anthropogenic and meteorological effects were in balance. These observations explain the apparent flattening reported in De Leeuw et al. (2023). After May 2021, the AOD started to decrease to a minimum in the summer of 2022, followed

by a rebound in 2023 (Fig. 5). More precisely, the monthly mean data in Fig. 4 show a strong decrease after a maximum in August 2021, to a low minimum in December 2021 followed by a period in 2022 when AOD was lower than in 2021 and 2023 and decreased between May and December.

The minimum in 2022 is attributed to anthropogenic conditions, meteorological effects were small and turned slightly unfavorable. The monthly mean AOD (Fig. 4) was much smaller than the simulated AOD, which peaked in July. The anthropogenic influences leading to the strong minimum in 2022, are anticipated to be a consequence of several COVID lock down periods and resulting economy slowdown in 2022 (Worldbank, 2022). Liu et al. (2024b) reported that the Shanghai lockdown in 2022 resulted in a stagnating economy in parts of China.

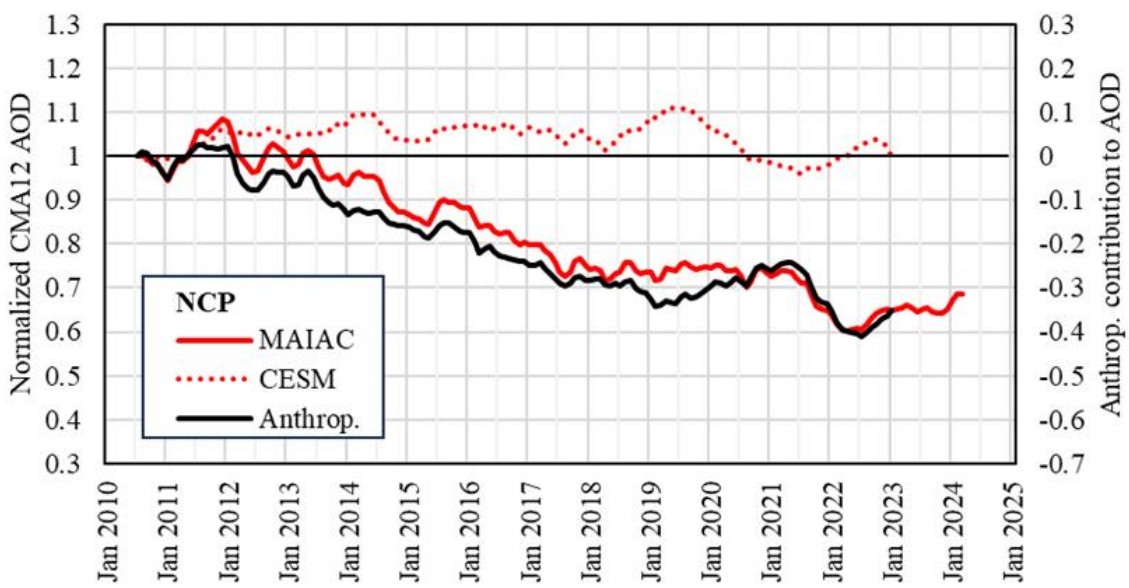

**Figure 5: Time series of CMA12 filtered MAIAC and CESM-simulated monthly mean AOD over the NCP from July 2010 to March 2024 and January 2023, respectively, normalized in July 2010. The black line shows the anthropogenic contribution to the AOD (secondary vertical axis). The thin black line has been drawn to guide the eye.**

### 3.3.2 AOD time series over the YRD

Time series of the monthly mean MAIAC and CESM-simulated AOD over the YRD are presented in Fig. 6. The simulated AOD time series show a regular pattern with peak values in March and minima in Oct/Nov of most years. The peak AOD values were around 0.65 and varied little during the first 6 years. In 2016 and 2017 the peak values decreased to 0.61 and 0.53, whereafter they gradually increased to 0.63 in 2021. The next 2 years they decreased to 0.53 in 2023. The simulated data show smaller secondary peaks between August and October, with varying intensities. Also, the minimum values vary from year to year, with the lowest value in 2017 and the highest value in 2014. The interannual variations in the simulated AOD show the clear influence of meteorological conditions which are illustrated by the variation in their effects on the AOD in Fig. 7. The meteorological effects were overall unfavorable, except in 2016 and 2017. Clear unfavorable conditions occurred in 2011 (up to 9%), 2014 (8%) and 2020 (8%).

The regular AOD pattern with maxima in March is also visible in the MAIAC data but with strong deviations in the first 5 years. In particular, anomalously high peaks occurred in June, in 2012 (AOD 0.97) and in 2014 (1.05), but also in 2010 and 2011 (both 0.86). In each of these four years, no clear peak was observed in March (as in the model simulations), the AOD was overall high, while relatively low minima occurred in December 2010 (0.31) and in July in 2012 and 2013 (both 0.32). During the first half of 2014 the AOD was overall higher than in other years but after the anomalously high peak in June, the AOD decreased during the second half of the year. From 2015, the observational data and the model simulations followed a similar pattern with maxima in March, often with another peak in June/July, and the observational peak AOD clearly decreased until 2018. The CMA12 time series shows that this decrease lasted until Spring 2018, followed by a substantial increase during about 1 year. From summer 2019 the AOD declined until August 2022, to a minimum AOD of 0.30 (CMA12 value). The monthly mean AOD confirms the strong decline in 2021/2022, to a minimum of 0.20 in October, followed by a rebound in 2023 with peaks in March and June with higher AOD than in 2021. The overall decrease of the AOD over the YRD between 2019 and 2022 is attributed to anthropogenic effects, with fluctuations due to meteorological effects. The lowest AOD was observed in 2022, decreasing between April and October. In contrast, the simulated AOD was substantially higher and peaked in August.

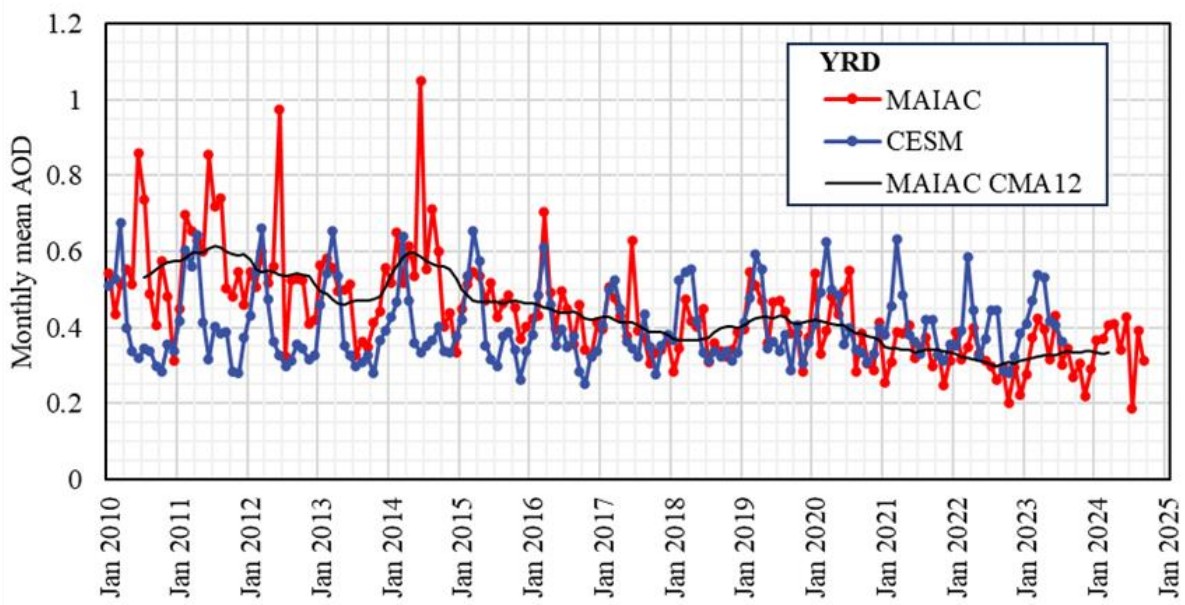

**Figure 6: As Fig. 4, but for the YRD.**

The normalized CMA12 time series in Fig.7 show an initial increase by 15% in the MAIAC AOD between mid-2010 and mid-2011, with similar contributions from unfavorable meteorological and anthropogenic effects. In 2012 the AOD was close to the July 2010 value and decreased by about 13% in 2013 before it increased to a maximum in April 2014. The maximum AOD was 12% larger than the July 2010 value, mainly due to anthropogenic effects, reinforced by a contribution from unfavorable meteorological conditions. The AOD remained high throughout 2014 but declined substantially by

17%, between October 2014 and April 2015, in spite of unfavorable meteorological conditions of about 7%. The AOD changed little during 2015 as a result of opposite anthropogenic and meteorological contributions. During the following years, 2016 - 2018, the meteorological contributions were very small while anthropogenic effects caused an increased reduction of the AOD, attributed to the 2013 - 2017 Clean Air Action Plan, resulting in a net decrease by 31% between July 2010 and early 2018.

This decrease was followed by an initial increase until mid-2019 (by ~12%, referenced to July 2010), whereafter the AOD steadily decreased further until mid-2022, which is mainly attributed to emission control policy as part of the 2018-2020 blue sky and the 14th five-year plans, with some variations due to unfavorable meteorological conditions of up to 6% in 2020. However, in the second half of 2021, the anthropogenic effects decreased but the increase in AOD was offset by meteorological effects which became less unfavorable. This resulted in a relatively constant AOD during the last months of 2021, followed by a strong decrease until mid-2022. The monthly mean AOD data in Fig. 6 show the contrasting behavior between the AOD in 2022 and both earlier and later years, in spite of the offset by meteorological effects as shown in both Figs. 6 and 7. Overall, a substantial AOD reduction was achieved between 2010 and 2022, to 57% of the 2010 value. The rebound in the second half of 2022 is suggested to be due to meteorological effects (Fig. 7). Lacking model data, the increase during 2023 cannot be further analyzed but is anticipated to be due the rebound of the economy and associated activities after the Shanghai lockdown (Liu et al., 2024b).

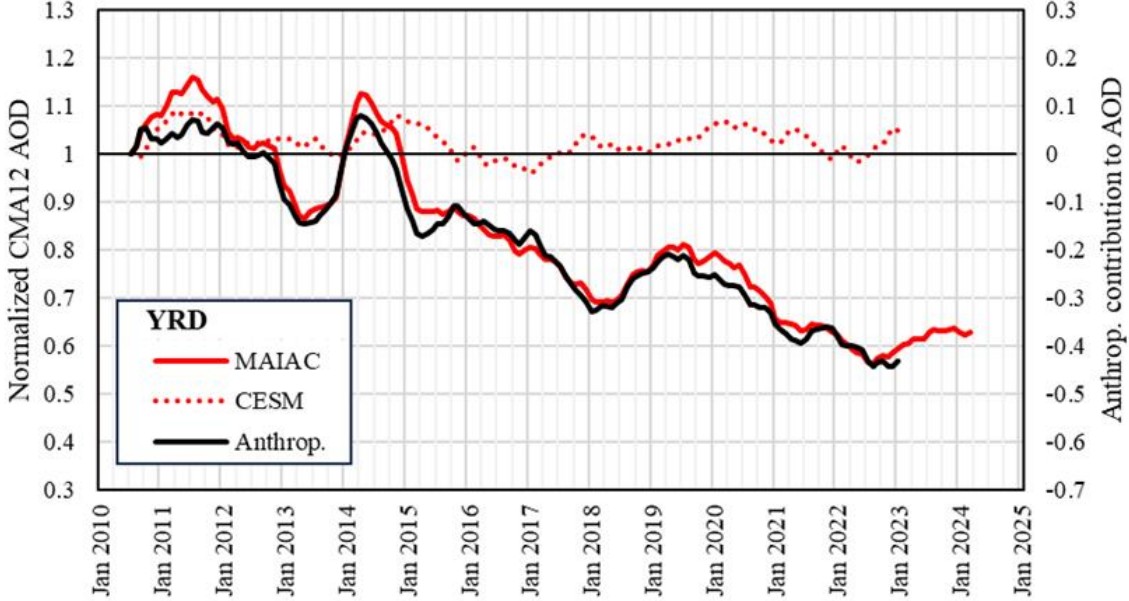

**Figure 7: As Fig. 5, but for the YRD.**

### 3.3.3 AOD time series over the PRD

Time series of the monthly mean MAIAC AOD and CESM-simulated AOD over the PRD are presented in Fig. 8. The time series of the simulated monthly mean AOD show a very regular pattern

with peak values in March/April and minima in July of most years. The simulated monthly mean AOD peak values increased substantially between 2011 and 2016, and between 2018 and 2021, with much lower values in 2017, 2018 and 2022. Overall, the peak AOD was higher during 2013 - 2016 than after 2016. The similar behavior of observed and simulated data indicates the influence of meteorological effects.

The peak values in the observational and simulated AOD data occur in about the same months, but the simulated maxima are much higher than those of the observations. A reason for this discrepancy may be the large influence of smoke on aerosols in the PRD (Zhang et al., 2010; Liu et al., 2021) which in the CESM model is treated as anthropogenic emissions and thus fixed at the 2010 level (Section 2.3). The data in Fig. 8 show that the simulated AOD in March 2010 was substantially higher than the observed AOD, suggesting that the initial anthropogenic emission estimates were high.

In-between the peaks, the simulations traced the observations reasonably well. In particular, the summer minima are well reproduced, whereas during the autumn the patterns are similar but the simulated AOD is higher. However, during the period June 2021 - February 2023, the simulated values show a clearly increasing tendency whereas the satellite-derived AOD was overall substantially smaller and in particular the maxima were much smaller than in other years. This clearly indicates that the emissions during this period were strongly reduced with respect to other years.

Secondary peaks are often observed in October, except in 2010 and 2011, shifted to September in later years and to August in 2019 and 2020. These peaks also occur in both data sets but are relatively larger in the observations than in the simulations.

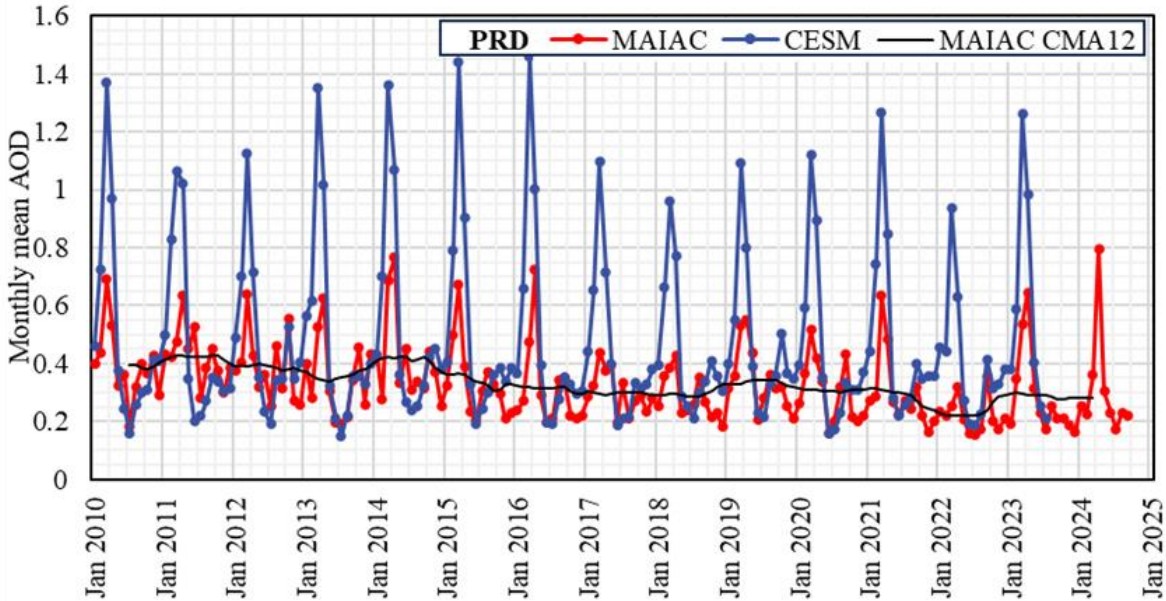

**Figure 8: As Fig. 4, but for the PRD**

Time series of normalized CMA12 of simulated and observational AOD data over the PRD are presented in Fig. 9, together with the anthropogenic contributions to the AOD. The model time series show that the meteorological effects on the AOD were initially favorable but anthropogenic effects

were large and unfavorable, which led to an increase in AOD by up to about 8% throughout 2011. In the first half of 2012, opposing meteorological and anthropogenic effects resulted in a net zero effect on the AOD. Anthropogenic effects changed to favorable which resulted in a minimum AOD in May 2013 (Fig. 9; the monthly mean data in Fig. 8 show a minimum in both the observational and simulated data in July). After May 2013, the anthropogenic effects gradually changed to unfavorable resulting in the high AOD observed throughout 2014. From September 2014, the anthropogenic effects changed to favorable and the AOD decreased during one year to a minimum in September 2015, whereafter the anthropogenic effects gradually became less favorable. Between May 2013 and mid-2016 the meteorological effects were small and fluctuated between favorable and unfavorable, thus modifying the AOD. However, from 2016 the meteorological effects were favorable with a reduction of the AOD of up to 14% in the winter of 2018. Favorable meteorological and anthropogenic effects of similar magnitude resulted in a gradual decrease of the AOD between the end of 2015 and mid-2018. Between July 2018 and July 2019, the AOD increased substantially due to both meteorological and anthropogenic effects changing to less favorable. From August 2019, the AOD decreased due to anthropogenic effects and in 2021/2022 a strong AOD minimum was observed which is attributed to meteorological effects, as indicated by both the monthly mean and filtered time series for the simulated AOD, reinforced by anthropogenic effects. The monthly mean observational data in Fig. 8 show that the AOD in the second half of 2021 and throughout 2022 was substantially lower than in 2020 and 2023, and likewise, the simulated AOD shows that in 2022 both the peak value in March and the summer minimum AOD were lower than in other years. In conclusion, the large AOD minimum in 2021/2022 was caused by contributions from both meteorological and anthropogenic effects between September 2021 and October 2022.

The CMA12 of the MAIAC observational AOD data show a broad maximum in 2014 which, as compared to the situation over the YRD, could not be caused by an anomalous peak in 2014 (compare Fig. 8 with Fig. 6). Rather, the monthly mean AOD was overall high in the second half of 2013 and in 2014, with month-to-month variations. The comparison with the centered moving average of the simulated AOD, normalized to 2010 (Fig. 9), shows that the AOD in 2011/2012 would have been substantially higher if it would not have been reduced by up to 10% due to favorable meteorological conditions.

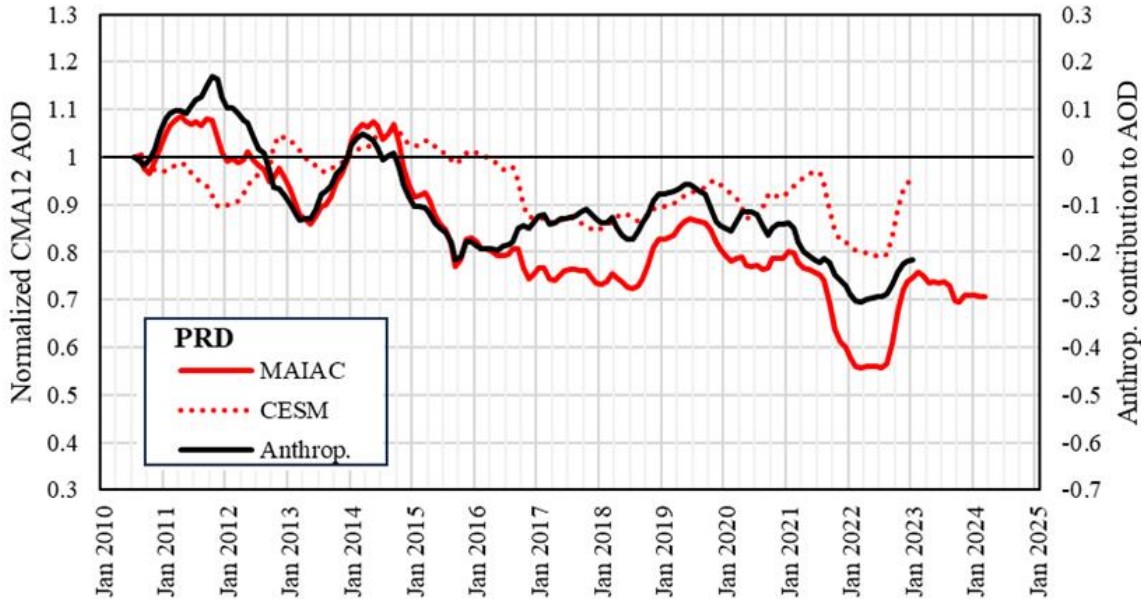

**Figure 9: As Fig. 5, but for the PRD.**

### 3.3.4 AOD time series over the HNB

The time series of the monthly mean model-simulated AOD over the HNB in Fig. 10 show a regular pattern with distinct peaks in March and minima in the summer centered around July. The simulated monthly mean AOD peak values vary by more than 15%: increase between 2011 and 2013, decrease between 2015 and 2018 when the simulated peak value is lowest, and increase between 2018 and 2020. Furthermore, in 2014 the minimum AOD was substantially higher than in other years and lasted a longer time, from May to December. Likewise, minima in 2018 and 2019 were higher and lasted longer than in other years. These variations in the simulated AOD show the large influence of the meteorological conditions on the AOD.

In contrast to the model simulations, there is no regular pattern in the monthly mean MAIAC observational AOD data. Three strong and distinct peaks were observed, all in June instead of March, in 2010 (0.93), in 2012 (0.90) and in 2014 (1.07). AOD maxima in March occur during the period 2015 - 2019, but overall, the variations in the observational data are not regular as in the simulated data. The CMA12 was high (~0.60) until February 2012 and again in early 2014, with a minimum in April 2013. This minimum reflects the clear decrease in the monthly mean AOD data from the secondary peak in March (0.61) to the lowest value in the first 5 years in July (0.27) to a maximum in October (0.57). The October maximum was the start of a period with high AOD leading into the 2014 peak in June, whereafter the AOD stayed relatively high until November. The 2013 minimum is quite well reproduced by the model simulations, except for the October peak. Furthermore, apart from the June peak values in the observational data in 2010, 2012 and 2014, the simulated AOD peak values are higher than those observed, which makes it hard to reach quantitative conclusions about the relative contributions from meteorological and anthropogenic factors from the comparison of monthly mean data.

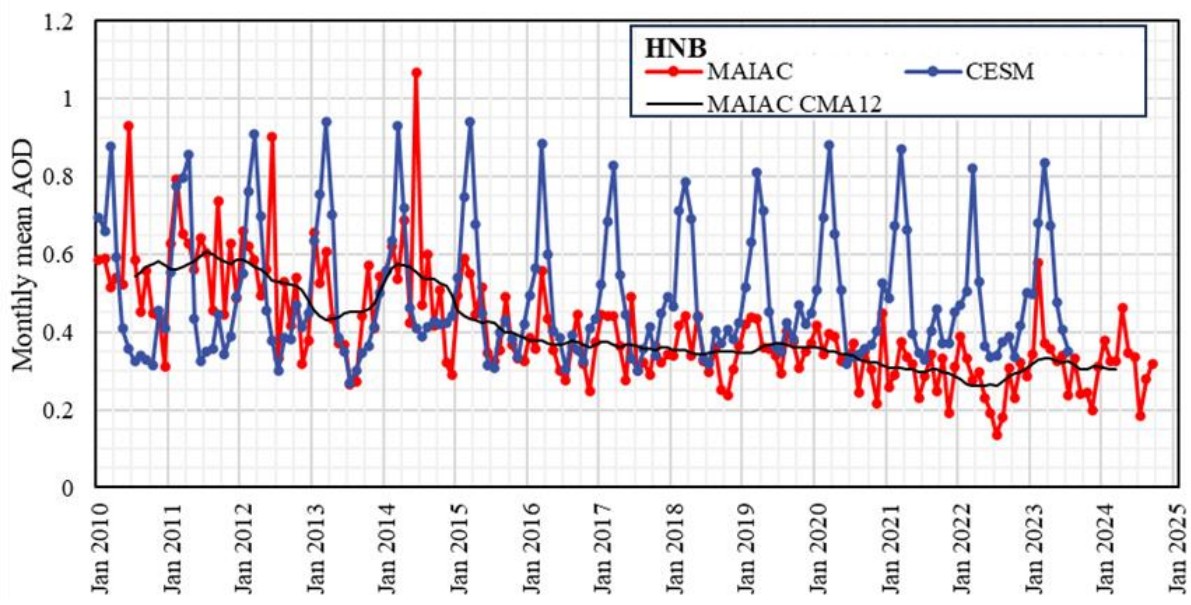


**Figure 10: As Fig. 4, but for the HNB.**
The normalized CMA12 time series of the model data in Fig. 11 show that the meteorological effects
on the AOD were unfavorable during the first 6 years (up to 10%) and much smaller thereafter,
fluctuating between favorable and unfavorable. The initially unfavorable meteorological contributions
resulted in an AOD increase in 2011, with only a small anthropogenic contribution. In the second half
of 2014, unfavorable meteorological effects were of similar magnitude as the favorable anthropogenic
effects, which resulted in a zero net effect on the AOD.
The AOD decreased overall from 2015 until the end of 2018, which was mostly attributable to
anthropogenic influences, the effect of which was modified by slowly changing meteorological effects
from slightly unfavorable to slightly favorable (Fig. 11). During 2019 the meteorological effects
changed to more unfavorable (up to about 7%) offsetting anthropogenic effects and thus causing a
small increase in the observed AOD. A relatively fast AOD decrease occurred between July 2019 and
the summer of 2021. During the second half of 2021 the anthropogenic effect became less favorable,
but the meteorological effect changed faster to more favorable, which effectively resulted in a small
AOD decrease. In early 2022 this was followed by a period when the monthly mean AOD (Fig. 10)
decreased from 0.39 in January to a minimum of 0.14 in July before it increased into a peak value of
0.58 in February 2023, the highest value since 2015. The low AOD in 2022 was in part due to the
favorable meteorological conditions while, as Fig. 11 shows, anthropogenic effects contributed to the
decrease in the first half of 2022. The anthropogenic and meteorological contributions were similar
during this period of low AOD. From August 2022 both effects were less favorable, leading to an
increase of the AOD.

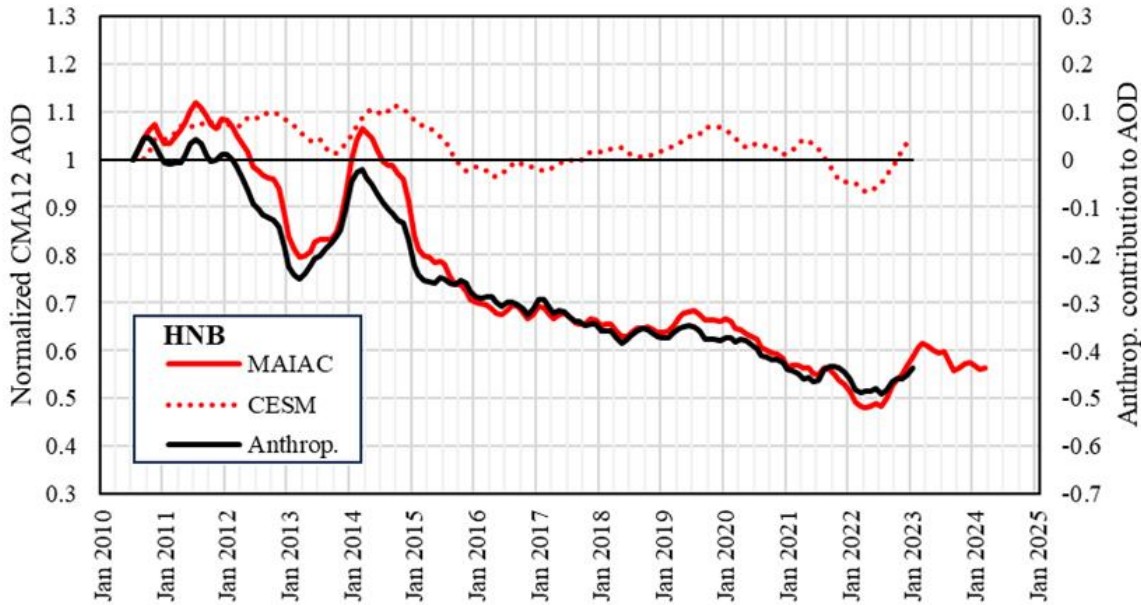


**3.3.5 AOD time series over the SCB**
Time series of the monthly mean MAIAC AOD and CESM-simulated AOD over the SCB are
presented in Fig. 12. The time series of the model-simulated AOD shows a regular pattern with distinct
peaks in March and minima in July and in most years a second minimum in September. The simulated
monthly mean AOD peak values are similar between 2010 and 2017, with some variation, whereas
from 2018 they are 10 - 16% lower. The simulated monthly mean AOD values are initially smaller
than the monthly mean MAIAC observational AOD values. In 2015 and 2016 the simulated and
observed peak values match well, and also in 2020, 2021 and 2023, with smaller observational values
in the intermediate years. The MAIAC AOD pattern was less regular than that of the simulations, with
distinct peaks visible in any of the winter months December - March, but also in June and September,
especially during the period 2010 - 2014 when AOD was high. Between 2018 and 2022 the MAIAC
AOD was lower, and peaked during the whole winter period.

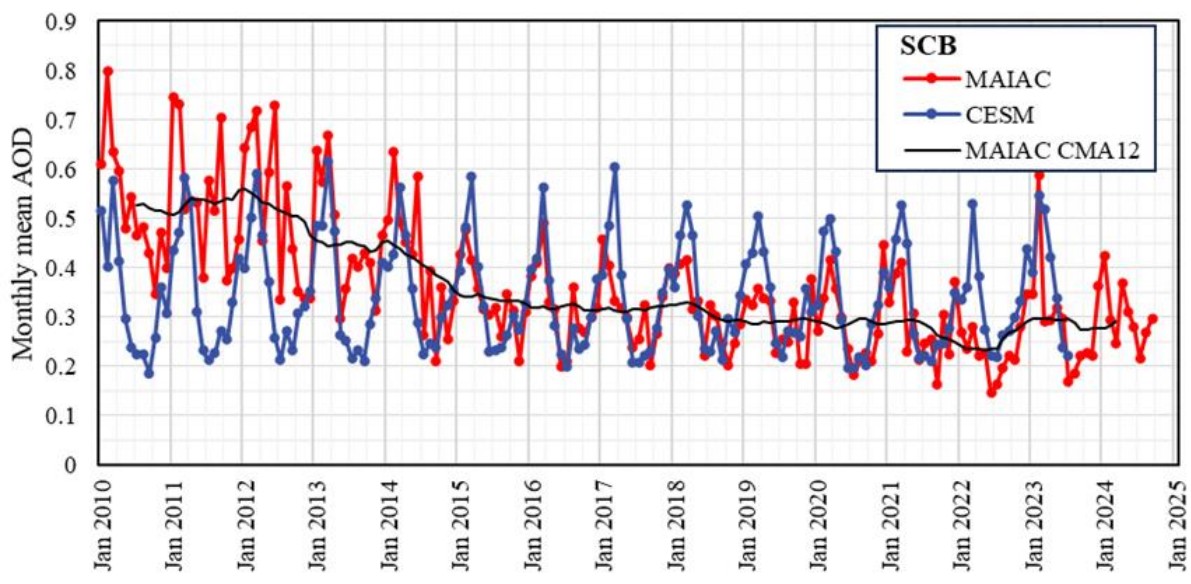


**Figure 12: As Fig 4, but for the SCB.**
Normalized CMA12 time series of the observational and simulated AOD over the SCB are presented
in Fig. 13. The model time series show that the meteorological effect on the AOD was up to 10%,
unfavorable during 2010 - 2015, small and mostly unfavorable between 2016 and 2020, favorable after
2020 (up to 9%) and again unfavorable from July 2022. The unfavorable meteorological effects offset
the favorable anthropogenic effects during 2010 and 2011, which resulted in small net changes of the
AOD and a maximum by the end of 2011. Accelerated emission reduction resulted in a fast decrease of
the AOD during the period 2012 - 2014, reinforced by the decrease in unfavorable meteorological
effects.
Between 2015 and 2019 a smaller reduction was achieved due to anthropogenic effects, with some
modifications attributed to meteorological influences, such as the small increase in early 2018 and the
decrease later that year. In contrast, during 2021 and 2022 the AOD decreased substantially to a
minimum in the summer of 2022. The monthly mean AOD data in Fig. 12 show the lower AOD
between May 2021 and the autumn of 2022, with a very low value in June 2022 (0.15). This decrease
was initially reinforced by favorable meteorological effects which were most effective in the first half
of 2022, as Fig. 13 shows. The rebound in the second half of 2022 was reinforced by unfavorable
meteorological effects.

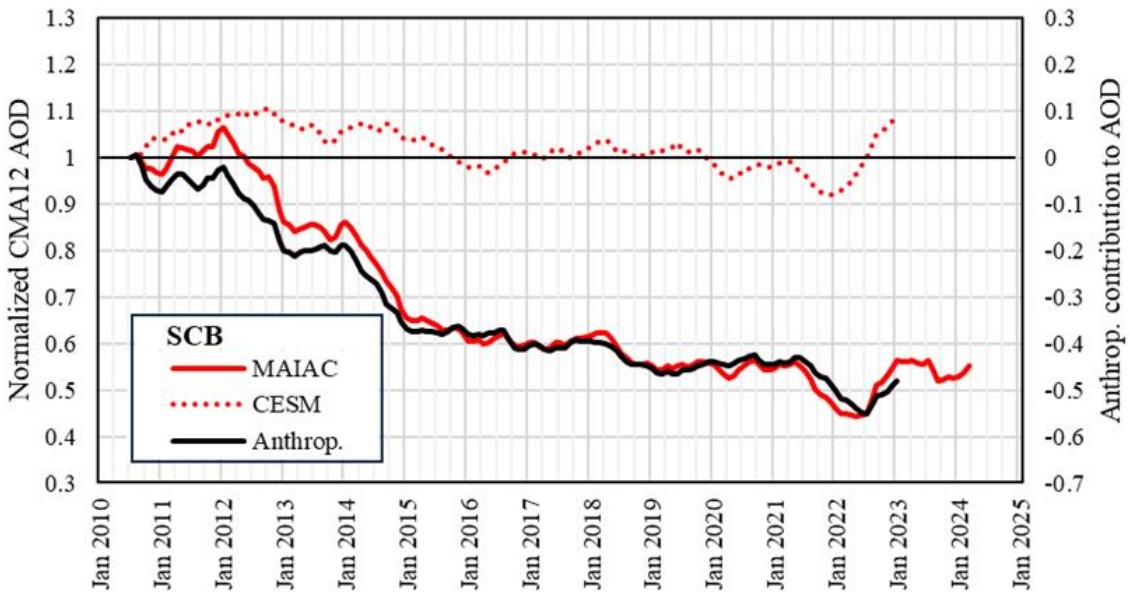


**Figure 13: As Fig. 5, but for the SCB.**

## 4 Discussion

In Section 3.3, the AOD time series were discussed for each of the five regions separately. Below, results common to two or more regions are discussed, for different periods of time with distinct features. Meteorological events influencing AOD variations are indicated but a detailed analysis is outside the scope of the current study and will be presented in a separate publication. In this study, both monthly mean and low-pass filtered monthly mean (CMA12) AOD data have been used. Monthly mean time series show more detail than the CMA12 time series as regards the start and end time of events affecting the evolution of the AOD. The monthly data points in CMA12 time series represent the average over 12 months around that data point and smoothed short term features. CMA12 time series clearly show tendencies which are difficult to determine in monthly mean time series due to the monthly and seasonal variations.

### 4.1 Overall effects of emission reduction policy on aerosol properties

Satellite measurements of AOD over China show that emission reduction policy has been successful in reducing the aerosol concentrations between 2010 and 2018, with an additional but smaller reduction toward the end of the study period, in 2024. Over the NCP, the AOD in 2024 had been reduced to 68% of its value in 2010, over the YRD to 62%, over the PRD to 70%, over the HNB to 55% and over the SCB to 57% (CMA12 values). In 2010 the AOD over the five regions ranged from 0.40 (PRD) to 0.53 (YRD and HNB), while in 2024 the AOD over the five regions ranged from 0.29 to 0.33 (see Figure 3). However, a closer look shows that the AOD did not vary monotonously and substantial variations occurred, as revealed after low-pass filtering (CMA12). The AOD not only varied between the five different regions, but the AOD variations within each region occurred at different times. Furthermore,

the monthly mean MAIAC and model AOD (Figs. 4, 6, 8, 10, 12) show that at the beginning of the study period (2010 - 2016), when the AOD was high, the patterns within a region were different from those at the end (2016 - 2024), when the AOD had substantially decreased. Before 2016, AOD peaks were observed over different areas and in different months than after 2016.

The high AOD peaks observed over northern, eastern and central China (Section 3.1) occurred mostly during the beginning of the study period, and those in June are attributable to emissions from agricultural straw burning (cf. (Liu et al., 2020a), for a map of straw burning locations). The intensive implementation of the ban on open crop straw burning between 2013 and 2018 resulted in a declining trend of $PM_{2.5}$ emissions in eastern and central China (Huang et al., 2021) which may explain why such high AOD maxima are not observed in June during later years. During these later years, elevated AOD peaks are observed over the PRD in the spring which are attributed to the transport of biomass burning plumes from Indochina during specific weather patterns (Xue et al., 2025).

The aerosol reduction, together with the occurrence of anomalous meteorological situations (Wang and He, 2015; Yin et al., 2024; Yin et al., 2017), may have resulted in changes in aerosol properties across different regions such as observed in, e.g., Shanghai (Wang et al., 2024) and Beijing (Li et al., 2021). In particular, variations in emissions of trace gases such as $SO_2$, $NO_2$ and organic vapors, which are aerosol precursor gases, influence aerosol composition (Zhang et al., 2015; Geng et al., 2017; Geng et al., 2019; Li et al., 2021). These authors show that the aerosol composition changed during the periods they studied. In particular, Jing et al. (2025) discuss the evolution of tropospheric aerosols over Wuhan during a period similar to that used in the current study (2010 - 2024). Using a comprehensive suite of data, Jing et al. (2025) indicate aerosol chemistry as a key factor for the evolution of aerosol properties and illustrate this with examples and case studies.

Aerosol chemistry influences the optical and physical properties of aerosols and thus may be a major factor leading to the different AOD patterns observed during the beginning and the end of the study period. The model simulations were made with fixed emissions and thus cannot reproduce this situation. In future studies, changes in aerosol properties in response to reduced emissions of precursor gases and associated chemical processes need to be taken into account.

**4.2 AOD reduction over different regions between 2010 and 2018 and influences of anomalous meteorological situations**

The AOD reduction in 2015 - 2018 can be attributed to the implementation of the 2013 - 2017 Clean Air Action Plan (Feng et al., 2019; Zhang et al., 2019c), but with variations due to meteorological effects, and different tendencies between the five regions. The AOD reduction achieved between 2015 and 2018 was smaller than that in earlier years, i.e. between the summers of 2011 and 2013, except over the NCP where the AOD decreased monotonously between December 2011 and May 2018. Over the SCB, most of the AOD reduction was achieved during two episodes between January 2012 and February 2015, in spite of unfavorable meteorological conditions and a year (2013) when the

anthropogenic reduction was discontinued. The accelerated decline over the SCB is attributed to the implementation of stricter emission standards for thermal power plants in 2012 which required all coal power plants to reduce effluent $NO_x$ (counted as $NO_2$) emissions to 100 mg/m³ or lower, except for some special unusual cases. This resulted in the quick installation of flue-gas denitration facilities such as Selective Catalytic Reduction equipment at coal-fired power plants (Mee and General Administration of Quality Supervision Inspection and Quarantine, 2011; Yan et al., 2023).

The interruption of the AOD decrease from the summer of 2013 until early 2015 is clearly visible in the monthly mean AOD data (Figure 2), showing elevated AOD between July 2013 and December 2014, over the YRD, HNB and PRD, with a much smaller AOD enhancement over the NCP. The interruption of the AOD reduction over the SCB in 2013 seems to be a combination of relatively high AOD in the summer/autumn of 2013 and the peak in January 2014, which in the CMA12 time series resulted in a flat AOD curve in 2013. The high AOD over the YRD, HNB and PRD resulted from elevated AOD, with respect to other years, in the second half of 2013 and most of 2014 as discussed in Section 3.3.2. This anomalously high AOD may be attributable to a combination of anomalous meteorological situations, such as winter haze in the NCP (Yin et al., 2017) and summer drought (Wang and He, 2015). By the end of this situation, in early 2015, the AOD was at a similar level as in the summer of 2013.

Yin et al. (2017) ascribed the occurrence of severe winter haze events in the North China Plain in 2014 to a weakened East Asian winter monsoon (EAWM) and anticyclonic circulation. Wang and He (2015) ascribed the North China / Severe Summer Drought in 2014 to a weakened East Asian summer monsoon (EASM). A weak EASM results in increased aerosol concentrations over northern China (Feng et al., 2016). Effects of El Niño–Southern Oscillation (ENSO) on air quality in southern China (i.e., south of the Yangtze River) were described by Wang et al. (2021): anticyclonic circulation during El Niño events weakens EASM resulting in low AOD. Vice versa, cyclonic circulation during La Niña events strengthens EASM resulting in high AOD. This may explain the stronger enhancement of the AOD in the PRD, YRD and HNB than in the NCP.

A similar situation as in 2014 may have occurred in 2011, when anomalous weather was reported during winter/spring (Sun and Yang, 2012; Jin et al., 2013). The monthly mean AOD data show that the AOD was high and meteorological effects were unfavorable. The CMA12 time series show that the AOD decreased between the summer of 2011 (YRD, HNB, PRD) or January 2012 (NCP, SCB) and 2013. The fast decrease of the AOD between the summers of 2011 and 2013 may thus have been due to a combination of meteorological effects and emission reduction in response to the implementation of environmental regulations. The AOD over the NCP and YRD was high in 2011 and both the meteorological and anthropogenic effects were unfavorable, but the latter changed to favorable in 2013.

**4.3 AOD variations after 2018 over different regions: increasing importance of meteorological influences**

The AOD observations and, in particular, the CMA12 time series, show that the AOD declined until 2018 over all regions, except over the SCB. However, from the summer of 2018, the AOD increased substantially during one year over the YRD, PRD and HNB, but not over the NCP and the SCB. The meteorological effects increased to less favorable over the PRD and to more unfavorable over the NCP, YRD and HNB. Thus, part of the AOD increase may be attributable to meteorological effects. However, over the NCP, YRD and PRD also the anthropogenic effects changed to less favorable during this period while a slight decrease was observed over the HNB and SCB, where the AOD enhancements were small (HNB) or did not occur (SCB). It is further noticed, that the increase of the meteorological effects to more unfavorable was substantially larger over the NCP than over other regions, but the anthropogenic effect continued to change to more favorable in 2018 while over the YRD and PRD it already changed to less favorable. Hence, the change in timing of the events and the magnitudes of the respective changes resulted in substantially different AOD variations. In particular, there was no AOD maximum over the NCP in 2019 due to the balance between anthropogenic and meteorological effects. Over the PRD both types of effects changed to more unfavorable, thus reinforcing each other. Over the YRD the very small meteorological effect caused minor AOD variations on top of strong anthropogenic effects. The interplay between variations of anthropogenic and meteorological effects influenced the effectiveness of the 2018 - 2020 three-year action plan for cleaner air.

In Section 4.1, the change in aerosol properties after 2016 was attributed to changing atmospheric composition in response to emission reductions indicating aerosol chemistry as a key factor, together with the occurrence of anomalous meteorological situations. Information on the latter is provided by the model simulations. As discussed in Section 3.3, the simulated monthly mean AOD time series show distinct patterns with a single maximum in a specific month, often accompanied by a secondary peak with lower AOD and a minimum covering a longer period. The patterns and peak months varied between regions and the maximum and minimum AOD values in each region varied from year to year. Distinct changes are observed in the variation of the peak values over the YRD and HNB in 2016, over the PRD in 2017, and over the SCB in 2018. These years mark the start of the period when peak AOD values were lower than in earlier years. The normalized CMA12 of simulated AOD time series in Section 3.3 show that over the YRD, HNB and PRD these years coincide with changes in meteorological effects, from favorable to neutral over the YRD and HNB and from neutral to favorable over the PRD. In each case these new situations persisted for several years. It is noted that an El Niño event occurred during 2015/2016 followed by La Niña events during 2016/2017 and 2017-2018 (Zhang et al., 2019a). Effects of El Niño and La Niña events on AOD were indicated in Section 4.2.

The above discussion shows the occurrence of substantial differences between the meteorological
effects over the YRD, PRD. HNB and SCB. Over the NCP, the meteorological effects were generally
unfavorable, resulting in a smaller AOD decrease between 2012 and 2019, with episodical increases in
2011, 2014 and 2015/2016, while during 2018 - 2020 the AOD did not change as a result of
cancellation of opposing effects of similar magnitude. These cases illustrate the increasing importance
of meteorological effects as the AOD becomes smaller: in 2014 and 2019 the meteorological effects
were similar but they were relatively more important in 2019.
Over the YRD, the meteorological effects were more variable than over the NCP, with stronger
unfavorable influences in 2011 and 2014/2015 when they reinforced unfavorable anthropogenic effects,
and during 2019 - 2022, when they reinforced strong anthropogenic effects, resulting in a strong
increase of the AOD. Over the PRD, the meteorological effects were mostly favorable, resulting in a
substantially lower AOD than due to anthropogenic effects alone. In particular the strong 2022
minimum benefitted from reinforcement by favorable meteorological effects. Over the HNB and the
SCB the meteorological effects were unfavorable during the first years of the study period, when they
reduced the effect of anthropogenic efforts, and small and variable thereafter. However, they did have
a favorable effect on the AOD during the 2022 minimum.

**4.4 Contribution of economic slowdown to the strong AOD decrease in 2022**

In contrast to the enhanced AOD events in 2011, 2014 and 2019 discussed above, a strong reduction
occurred in 2022, which over the YRD and HNB were preceded by a small anomaly in 2021. As
described in Section 3.3, the strong reduction in 2022 was clearly observed in the monthly mean data,
and mainly attributed to anthropogenic effects with a small contribution from meteorological effects.
The year 2022 was anomalously warm with heat waves and drought, with a specific meteorological
situation (Xu et al., 2024), including an extended La Niña event that persisted from 2020 to 2023
(Iwakiri et al., 2023), which may have influenced the aerosol properties in a variety of ways. Changes
in large scale circulation may affect transport pathways and the evolution of the atmospheric boundary
layer influences local and regional transport as well as ventilation, changes in relative humidity
influence aerosol optical properties, and air temperature influences the formation of new aerosol
particles. The occurrence of La Niña may have caused an increase in aerosol concentrations, whereas,
during heat wave events the AOD is reduced with downstream regions experiencing increased AOD
(Tseng et al., 2024). However, the 2022 minimum is mainly attributable to anthropogenic influences,
i.e. the economy slow-down  (Liu et al., 2024b) resulting in reduced production and transport across a
large part of China when sea harbors were closed and export stagnated (Wang and Su, 2025). As
discussed in Section 3.3, favorable meteorological effects contributed to reinforce this minimum in the
SCB, HNB and especially the PRD.

**5 Conclusions**

Satellite-derived monthly mean time series of AOD over five representative regions in China were presented for the period January 2010 - September 2024. AOD variations, both temporal variations within a region and differences between the five regions, were discussed and contributions from anthropogenic and meteorological factors were analyzed based on comparison with model simulations. The time series confirm the effective reduction of the AOD over China, which is attributed to the implementation of a series of policy measures aimed at the reduction of emissions of aerosols and trace gases to improve air quality. However, meteorological effects have a large influence on the aerosol load over China and their importance increases as AOD decreases. In particular in recent years, the important role of meteorological factors has become evident, for instance from the large increase of the AOD over the YRD, and to a smaller extent over the HNB and PRD, between 2018 and 2021. During this period, the potential decrease of the AOD over the NCP was effectively cancelled due to unfavorable meteorological effects. The results presented lead to the following conclusions:

- Emission reduction policy has resulted in the effective decrease of the AOD over China, in particular between 2015 and 2018 in response to the 2013 - 2017 Clean Air Action Plan. However, the 2018 - 2020 three-year action plan for cleaner air and the 14th five-year plan have been less effective: the overall reduction of AOD between 2018 and 2024 was substantially smaller than in the previous period.

- The effectiveness of the anthropogenic effects initially decreased during different periods of 1 - 2 years between 2018 and 2024, over the NCP, YRD and PRD, followed by a stronger AOD reduction until mid-2022 and an increase in January 2023 to a level that was similar to that by the end of 2021. This resulted in an AOD in January 2023 that was reduced with respect to that in 2018, due to only anthropogenic effects. This reduction is attributed to emission reduction which, however, was smaller than in 2013 - 2017. The strong AOD decrease in 2022 is shown as an anthropogenic effect, but did not happen in response to emission reduction policy. Rather, this is a short term effect that is mainly attributed to the "Shanghai" lock down effect (Liu et al., 2024b) which resulted in stagnation of the economy in large parts of China and associated activities such as transport and energy production. The accelerated decrease of the AOD over the SCB before 2015 is attributed to the implementation of strict emission standards for thermal power plants in 2012.

- Meteorological effects have a substantial influence on the AOD, which increased as AOD decreased. Unfavorable meteorological effects have been shown to reduce effects of emission reduction on AOD and favorable meteorological effects can reinforce emission reduction effects. Meteorological effects contributed to the AOD increase over several regions in 2018 and 2019 and affected the decrease in 2022, thus counteracting emission reduction measures

implemented as part of the 2018 - 2020 three-year action plan for cleaner air and the 14[th] five-
847         year plan during the period 2018 - 2024.

-   Very high AOD was observed in 2011 and 2014 which may in part be caused by anomalous
meteorological situations. The high AOD in 2014 has been suggested to be due to anomalous
circulation associated with effects of El Niño / La Niña and the strengths of the East Asian
summer and winter monsoon. These effects have become increasingly important in recent
852         years and are a major reason for the slowdown of the AOD reduction since 2018.

-   AOD variations show distinctly different patterns during the periods before and after 2016,
suggested to be due to changes in aerosol composition and optical and physical properties in
response to the reduction of aerosol precursor gases.

The results were obtained using new experimental data, i.e., the new MAIAC C6.1 AOD data that
replace and improve upon MAIAC C6 and extend the time series beyond 2021. The extended time
series reveal new phenomena and new insights as explained in detail in Section 3. The discussion in
Section 4 focused on features common to the five areas, as well as differences across them. These new
phenomena include the substantial increase around 2019 over the YRD, the PRD and, to a lesser extent,
over the HNB, a clear minimum in 2022 over all areas with a strong recovery in 2023, decreasingly
favorable anthropogenic contributions over the NCP in 2019 - 2021, over the YRD in 2018 - 2019 and
the stagnation of the AOD decrease over the PRD due to the declining favorable contributions between
2016 and 2019. Furthermore, the comparison between the AOD time series between 2010 and 2016
and in later years shows different patterns between these periods, both in the observations and the
simulations. In view of the length of the current MS, the detailed analysis and discussion of these new
phenomena will be presented in a separate publication.
The objectives stated in the Introduction were addressed throughout this paper. Data presented and
discussed show that the flattening of the AOD reduction between 2017 and 2021 suggested by De
Leeuw et al. (2023) was a consequence of the offset of AOD reduction by unfavorable meteorological
effects (Sections 3.3.1 and 4.3) (Objective 1), as also observed during earlier periods. The anomalous
AOD in the winter of 2014 over the YRD, HNB and PRD has been explained by large scale
meteorological effects influencing AOD, in particular by ENSO and East Asian winter and summer
monsoon (Section 4.2) (Objective 2). Relations between anomalous AOD and meteorological
situations have been discussed in Sections 4.2, 4.3 and 4.4 (Objective 4). Changing AOD patterns have
been reported and suggested to be due to changing atmospheric composition in response to selective
emission reduction policy (Section 4.1), together with the occurrence of anomalous meteorological
situations (Section 4.3) (Objective 5). Monthly mean AOD data were used throughout the paper to
identify the occurrence of specific events (Objective 3).
In summary, emission reduction policy has been effective in reducing AOD, but with many deviations
due to meteorological effects. Phenomena such as ENSO, East Asian winter and summer monsoon and
Heat Waves have been reported to influence AOD and $PM_{2.5}$, through their effects on large scale
circulation, regional transport and local meteorology. As discussed in Section 1, AOD and PM$_{2.5}$ are
different measures for aerosol concentrations but are not comparable. Qi et al. (2022) report that
influences of meteorology changes on AOD trends are larger than those on surface PM$_{2.5}$. Hence,
findings of meteorological effects on AOD reported in the current study may be significantly different
from findings from studies on meteorological effects on PM$_{2.5}$. The importance of the current study is
the use of AOD in climate studies on interaction with solar radiation for instance for meteorological
purposes, such as effects on heat waves (Wu et al., 2021) and local meteorology (Zhang et al., 2018),
or application in solar energy (Lin et al., 2023).
**Acknowledgments**
This work is supported by the National Natural Science Foundation of China [grant number
42305151); the National Key R&D Program of China [grant number 2023YFB3907405]; and the
President's International Fellowship Initiative of the Chinese Academy of Sciences [grant number
2025PVA0014]. The study contributes to the ESA/MOST cooperation project Dragon 6, Topic "Air
Quality Monitoring and Analysis in Populous areas in China (AQMAP).
**Author contributions**
All authors contributed to the study's conception and design. All authors read, reviewed and approved
the final manuscript. The study was designed and discussed by GdL, CFan, XY and HK. CFan and JD
collected data. HK and Fang contributed to the modeling. CFan, JD, GdL and XY wrote the first
paper draft, which was finalized by contributions from ZL, YZ, HK and Fang.
**Code/Data availability**
The data can be accessed by contacting the corresponding authors. MAIAC data are freely available
from the NASA's Land Processes Distributed Active Archive Center (https://lpdaac.usgs.gov/data/).
**Competing interests**
The authors declare that they have no conflict of interest.

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
