# Peer review of "Evolution of aerosol optical depth over China in 2010-2024"

_EGUsphere, 2025_

## Author Comment (AC1)

**Response to RC1**: 'Comment on egusphere-2025-880', Anonymous Referee #3, 14 June 2025

On behalf of all co-authors, we thank Referee #3 for the insightful and extensive comments which certainly contribute to the substantial improvement of the manuscript (MS). Below we respond to each of the specific comments which are copied below (in black). After each comment we provide our response, in red, together with changes in the revised MS. Line numbers (indicated by L) mentioned by Referee #3 refer to the original MS as published in the ACP discussion Section and revisions are quoted with line numbers (indicated by LR) referring to the revised MS.

The manuscript carefully assessed and determined the contributions of meteorological and anthropogenic effects on aerosol variations on monthly and inter-annual scales of China by taking five key areas and representations based on time series analysis of MODIS/MAIAC C6.1 AOD product and CESM model. It highlighted the complex interplay between meteorological and anthropogenic factors in shaping AOD variations across China and confirmed the increasing significance of meteorological effects in shaping China's AOD. Overall, the paper is well-written and presents novel findings. However, this manuscript needs some revisions before publication.

Thank you for these kind comments.

Specific comments:

1. The evolution of aerosol optical depth over China or East China or typical region, should be clearly indicated in title and abstract of the manuscript.

Thank you for this comment. This information is included in the title and abstract: The title starts with "Evolution of aerosol optical depth over China" (L 1-2) and the abstract starts with "Time series of MODIS/MAIAC C6.1 aerosol optical depth (AOD) over China were used together with model simulations of AOD to determine contributions of meteorological and anthropogenic effects on aerosol variations on monthly and interannual scales" (lines 26-28).

2. Line 47, "NCP, YRD, PRD, HNB and SCB", Abbreviations should be given their full names when they first appear.

Thank you for this comment. However, the word limit for the abstract is 250 words, which does not leave much room for more detailed information. We preferred to provide more information on the content of the MS, and therefore chose to provide the full names in the MS, Sect. 1 (lines 126-129).

3. Line 47, "An aerosol" may be modified as "Aerosol".

Thank you for this comment. We have checked and "An Aerosol" (L 48; LR 48) is correct use of English language.

4. Line 50, Pay attention to the font size of the word "atmosphere".

Thank you for this comment. We have corrected the font size (L 51; LR 51).

5. Figure 1 (line 190-191) shows mainly south-east China, is it suitable to use Southeast Asia other than south-east China for Figure 1's title? Please confirm.

Thank you for this comment. As advised by the Editorial staff, we have changed the Figure but this was not included in the MS as we were advised to do that later, after first revision.

6. Please provide the full name at abbreviations' first appearance to ensure clarity for readers who may not be familiar with these terms, such as MODIS (line 96), ENSO (line 732), etc.

Thank you for this comment. We have added the full names at first occurrences (LR 99), (LR 715), etc.

7. Line 201, MODIS sensor also has two channels with a pixel size of 250 m at nadir.

Thank you for this comment. We have changed the sentence to "a nominal pixel resolution at nadir of 250m (2 bands), 500m (5 bands) and 1000m (29 bands)," (LR 226).

8. Line 227-232, some studies show MAIAC AOD have a good validation accuracy in East China (the study area), which can be added in the description.

Thank you for this comment. We have modified the text in Section 2.3 and added more information on the validation of MAIAC C6.1: "MAIAC C6.1 has been validated over China by Ji et al. (2024) and Huang et al. (2024). Both studies report that the overall accuracy of the MAIAC AOD products over China is good. The validation by Ji et al. (2024) over bright surfaces, using publicly available reference data from AERONET and CARSNET until 2014, shows a significant underestimation and negative bias of the MAIAC C6.1 product, which however performs slightly better than DB and C6. The comparison with collocated AERONET AOD data, for the period from 2001 to 2021, by Huang et al. (2024) shows good consistency, with correlation coefficients (R) of 0.933/0.939, root mean square error (RMSE) of 0.152/0.146, bias of 0.005/0.015, mean absolute error (MAE) of 0.094/0.092, relative mean bias (RMB) of 1.221/1.301 and percentage of data points within expected error (EE) of 71.02/68.36. These statistical metrics refer to comparison at the overpass times of the Aqua (13:30 LT) and Terra (10:30 satellites, respectively (Huang et al., 2024, Fig. 2). The comparison shows a slight overestimation of C6.1 at low AOD (<0.5) and a small underestimation at higher AOD." (LR 252-263).

9. The line color of SCB in Figure 2 (line 272), Figure 3 (line 303) is not quite clear, use a darker color?

Thank you for this comment. We have used the color scheme recommended by ACP.

10. Line 296, Pay attention to the font size of the word "tendencies", "in different regions" should be modified as "in all/each regions".

Thank you for this comment. We have changed the font size of "tendencies". As regards the second comment, we have changed the words to "AOD reduction in the five regions".

11. Line 597, what does "These authors" referring to?

Thank you for this comment. We have provided a number of references in the previous sentence (613-614). "These authors" referred to the first authors and their co-authors of these papers. However, this was written in Section 4 and Section 4 has been re-written and re-organized for better and more clear presentation of the results. Now this text appears in Section 4.1 (LR 675-676).

12. Line 614-615, cite the reference "MEE & General Administration of Quality Supervision Inspection and Quarantine, 2011", and use the correct citation format of Yan et al. (2023).

Thank you for this comment. We have added the citation for this reference and provide the correct citation format for Yan et al. (2023) in the text and the list of references (LR 699-700).

---

## Author Comment (AC2)

**RC3**: 'Comment on egusphere-2025-880', Anonymous Referee #4, 14 June 2025

On behalf of all co-authors, we thank Anonymous Referee #4 for the insightful and extensive comments which certainly contribute to the substantial improvement of the manuscript (MS). Below we respond to each of the general and specific comments which are copied below in black. The general comments consist of two parts which we numbered as GC1 and GC2, where GC1 provides a brief summary of our study and GC2 provides a summary of suggestions which are further elaborated in the specific comments. We have numbered the specific comments as SC1-SC6. After each comment we provide our response, in red, together with changes in the revised MS. Line numbers indicated by L, mentioned by Referee #4, refer to the original MS as published in the ACP discussion section and revisions are quoted with line numbers (indicated by LR) referring to the revised MS.

General Comments

**GC1:** This manuscript presents an updated analysis of aerosol optical depth (AOD) trends over China from 2010 to 2024, using MODIS/MAIAC C6.1 satellite data and CESM/CAM5 model simulations to distinguish between anthropogenic and meteorological influences. The geographic scope and time span are appropriate, and the study makes effective use of established modeling techniques.

**Response to GC1**: Thank you for this summary. However, we would like to clarify that the AOD dataset from 2010-2021 in de Leeuw et al. (2023) was MAIAC C6. In this study, the 2010-2021 MAIAC C6 has been replaced with MAIAC C6.1 and then extended until September 2024. This implies that the whole AOD data set is C6.1, hence new data are used which are not merely an extension of the C6 data set used by de Leeuw et al. (2023). In addition, MAIAC C6.1 AOD data differ from C6 AOD data in several aspects (see below). We have reformulated the text to "the C6 time series for 2010-2021 used by de Leeuw et al. (2023) was replaced with the recently released (6 July 2022) MODIS/MAIAC C6.1 data and extended with C6.1 data until September 2024." (LR 142-143). Hence a new data set has been used for the whole study period. MAIAC C6.1 is different from MAIAC C6 (L 139-143), (LR 144-147).

Differences between C6 and C6.1 have been published in papers by Lyapustin and Wang (2022), LPDAAC (2024), Ji et al. (2024), Huang et al. (2024). We briefly describe these differences in Section 2.2. We have also addressed the differences between C6.1 and C6 with a focus on the 5 areas used in our study (until 2024), and in more recent years than used by Li et al. (until 2014) and Huang et al. (until 2021). However, including the analysis of the differences between C6.1 and C6 would render the MS very long and result in the loss of focus. Therefore, we decided to remove this analysis and use the results to prepare a separate paper where we also include validation results using other datasets (SONET), for other years and other areas than Li et al. and Huang et al. We have added a reference to this work "(Fan et al., 2025; in preparation)" (LR 147 in the revised version). An extensive description of these differences is beyond the scope of the current paper, since we do not make specific comparisons with de Leeuw et al (2023), for reasons mentioned in L 135-147).

The validation by Ji et al. (2024) was made over bright surfaces, which are outside our five study areas, and includes data which were mostly collected before our study period. Hence these results have general relevance for the assessment of the accuracy of MAIAC C6.1 AOD data (and therefore this reference is included) but are not relevant for our study. The validation by Huang et al. (2024) includes data in our study areas and overlaps in time (until 2021) with most of our study period. We have included the statistical metrics provided by Huang et al. (2024) in the revised MS and the paragraph now reads: "MAIAC C6.1 has been validated over China by Ji et al. (2024) and Huang et al. (2024). Both studies report that the overall accuracy of the MAIAC AOD products over China is good. The validation by Ji et al. (2024) over bright surfaces, using publicly available reference data from

AERONET and CARSNET until 2014, shows a significant underestimation and negative bias of the MAIAC C6.1 product, which however performs slightly better than DB and C6. The comparison with collocated AERONET AOD data, for the period from 2001 to 2021, by Huang et al. (2024) shows good consistency, with correlation coefficients (R) of 0.933/0.939, root mean square error (RMSE) of 0.152/0.146, bias of 0.005/0.015, mean absolute error (MAE) of 0.094/0.092, relative mean bias (RMB) of 1.221/1.301 and percentage of data points within expected error (EE) of 71.02/68.36. These statistical metrics refer to comparison at the overpass times of the Aqua (13:30 LT) and Terra (10:30 LT) satellites, respectively (Huang et al., 2024, Fig. 2). The comparison shows a slight overestimation of C6.1 at low AOD (<0.5) and a small underestimation at higher AOD." (LR 252-263).

**GC2:** However, while the extension of the dataset is valuable, the current manuscript lacks scientific rigor in several sections. Most importantly, the paper does not frame its analysis with clearly stated hypotheses or research questions, making it difficult to evaluate the strength of its conclusions. It also fails to provide a rigorous statistical treatment of the data: there is no formal trend analysis, no uncertainty assessment, and no sensitivity testing of model assumptions. These limitations undermine the robustness of the findings and should be addressed in the new version. Additionally, the manuscript would benefit from improved transparency and stronger engagement with regional climate phenomena (e.g., ENSO, East Asian monsoon) which are only superficially mentioned in the discussion.

**Response to GC2**: GC2 is a summary of comments and suggestions which are further explained in the specific comments SC1 (hypotheses or research questions), SC2 (statistical treatment of the data: there is no formal trend analysis, no uncertainty assessment), SC3 (model assumptions) and SC6 (engagement with regional climate phenomena (e.g., ENSO, East Asian monsoon)). Therefore, we provide our responses to GC2 below each specific comment SC1-SC6.

**Specific Comments**

**SC1:** Lack of Hypothesis or Research Questions: the paper would benefit from explicitly stating research hypotheses. Currently, the study lists trends without anchoring them to hypothesis. For example: "Has the relative influence of meteorology increased over time?" or "Does the CESM model successfully replicate observed regional AOD variability under fixed emissions?"

**Response to SC1**: Thank you for these comments, which made us realize that we should more clearly state the research questions which in the submitted version were hidden in the text (L 148, L 166 and L 163). We have added the following text at the end of the Introduction (LR 173-180). "The objectives of the current study are (1) to investigate the reasons for the flattening of the AOD reduction during 2017-2021, observed by De Leeuw et al. (2023); (2) to investigate what caused the anomalous AOD in the winter of 2014 over the YRD, HNB and PRD, but not over the NCP and SCB (De Leeuw et al., 2023; Fig. 7); (3) to use monthly mean AOD data to accurately identify the start and end of anomalous events, which are hidden in the low-pass filtered data used in De Leeuw et al. (2023); (4) to connect the occurrences of anomalous AOD to specific meteorological conditions and/or anthropogenic interferences; (5) to investigate whether changes in aerosol physicochemical characteristics, in response to emission reduction and climate change, results in different AOD patterns.

Obviously, not all of these questions can be fully addressed in a single study. In the current paper we report and describe the observational data and provide comparisons with the CESM model data (with emissions fixed in 2010). Meteorological and anthropogenic effects on the AOD variations are discussed and possible influences of large scale meteorological effects (El Niño, La Niña, heat waves) and anthropogenic effects (policy measures and economic effects) are indicated. These effects will be discussed in more detail in a follow-up paper. "

These five objectives have been addressed throughout the paper and were discussed ias indicated in the following text added to Section 5: "The objectives stated in the Introduction were addressed throughout this paper. Data presented and discussed show that the flattening of the AOD reduction between 2017 and 2021 suggested by De Leeuw et al. (2023) was a consequence of the offset of AOD reduction by unfavorable meteorological effects (Sections 3.3.1 and 4.3) (Objective 1), as also observed during earlier periods. The anomalous AOD in the winter of 2014 over the YRD, HNB and PRD has been explained by large scale meteorological effects influencing AOD, in particular by ENSO and East Asian winter and summer monsoon (Section 4.2) (Objective 2). Relations between anomalous AOD and meteorological situations have been discussed in Sections 4.2, 4.3 and 4.4 (Objective 4). Changing AOD patterns have been reported and suggested to be due to changing atmospheric composition in response to selective emission reduction policy (Section 4.1), together with the occurrence of anomalous meteorological situations (Section 4.3) (Objective 5). Monthly mean AOD data were used throughout the paper to identify the occurrence of specific events (Objective 3)." (LR 852-863).

As regards the second proposed hypothesis "Does the CESM model successfully replicate observed regional AOD variability under fixed emissions?", this is not a research question because the emissions are fixed in 2010 ("To isolate the effects of meteorology on AOD, anthropogenic emissions were fixed using the monthly values from 2010, which were repeatedly applied to the corresponding months of each subsequent year." (LR 273-274) and therefore cannot be expected to reproduce the AOD variability in an environment where emissions are changing. "However, as discussed in Kang et al. (2019), the CESM results are not representative for actual situations because AOD was simulated using fixed emissions to identify meteorological effects on the AOD. The CMA12 filtered observational and simulated AOD data were normalized for quantitative comparison and to determine anthropogenic and meteorological influences, as explained in de Leeuw et al. (2023)" (L397-401; LR 356-360). The comparisons between (normalized) CESM simulated and observation data are discussed in Section 3.3.1. – 3.3.5.

**SC2:** Statistical Rigor and Trend Analysis: while normalized CMA12 filtering is used, the paper does not conduct any formal trend tests (e.g., Mann-Kendall, Sen's slope). It is missing the uncertainty assessment, confidence interval, or error propagation. As a result, it is unclear whether observed differences (e.g., 2014 vs. 2018 AOD) are statistically significant.

**Response to SC2**: Thank you for this comment. Long term trends were determined and discussed in De Leeuw et al. (2023), using rigorously ((KZ(12,3)) filtered AOD time series. The less rigorously filtered data in Fig. 3 of the current MS show that there are many fluctuations and anomalies while in most regions (except NCP) there is no period long enough to justify the determination of a trend. Hence, we have not attempted any trend analysis in the current study where we focus on anomalies and identification of events that may cause them (see also **Response to SC1**).

Having mentioned this, we do notice a general decrease of the AOD in the monthly mean AOD time series (Fig. 2) and more clearly in the CMA(12) filtered time series in Fig.3. We address the decrease in the five regions in the revised version in the Discussion (Section 4.1): "Satellite measurements of AOD over China show that emission reduction policy has been successful in reducing the aerosol concentrations between 2010 and 2018, with an additional but smaller reduction toward the end of the study period, in 2024. Over the NCP, the AOD in 2024 had been reduced to 68% of its value in 2010, over the YRD to 62%, over the PRD to 70%, over the HNB to 55% and over the SCB to 57% (CMA12 values). In 2010 the AOD over the five regions ranged from 0.40 (PRD) to 0.53 (YRD and HNB), while in 2024 the AOD over the five regions ranged from 0.29 to 0.33 (see Figure 3)." (LR 650-655)

As regards uncertainties, we refer to the validation by Ji et al. (2024) and Huang et al. (2024), as well as our own work (Fan et al., in preparation), all versus reference data sets. See our **Response to GC1** for more detail.

**SC3:** Model Evaluation and Bias Quantification: the CESM model simulations are presented as a basis for attribution to meteorological effects, yet no validation of model performance is provided beyond qualitative agreement. A quantitative comparison between model and satellite AOD (e.g., RMSE, bias, $R^2$) across regions would strengthen credibility.

**Response to SC3**: Thank you for this comment. However, as mentioned in the MS, "Model performance on aerosol has been widely evaluated (Lamarque et al., 2012; Fang et al., 2020; Emmons et al., 2010)" (L246-247). In the current study, "To isolate the effects of meteorology on AOD, anthropogenic emissions were fixed using the monthly values from 2010, which were repeatedly applied to the corresponding months of each subsequent year. Meteorological input fields, including horizontal winds, air temperature, surface pressure, land surface temperature, heat fluxes, and wind stresses, were nudged to the MERRA-2 (Modern Era Retrospective analysis for Research and Applications, Version 2) reanalysis dataset (Gelaro et al., 2017; Rienecker et al., 2011) (see also https://rda.ucar.edu/datasets/d313003/; last access 4 March 2024), which provides data at a 3-hour temporal resolution." (LR 273-279). As mentioned in our **Response to SC1**: the model cannot be expected to reproduce the AOD variability in an environment where emissions are changing. "However, as discussed in Kang et al. (2019), the CESM results are not representative for actual situations because AOD was simulated using fixed emissions to identify meteorological effects on the AOD. The CMA12 filtered observational and simulated AOD data were normalized for quantitative comparison and to determine anthropogenic and meteorological influences" (L319-321). The comparisons between (normalized) CESM simulated and observation data are discussed in Sections 3.3.1. – 3.3.5.

**SC4:** Sensitivity and Robustness Checks Missing: the fixed-emissions approach assumes that changes in modeled AOD are solely due to meteorological variability. However, no sensitivity analyses are provided to test this assumption. Could other fixed assumptions (e.g., emissions inventory resolution, nudging method) bias the attribution?

**Response to SC4**: Thank you for this comment. Indeed, in our research we followed the fixed-emissions approach, assuming that changes in modeled AOD are solely due to meteorological variability because meteorological parameters are the only ones that change during each month after the first year (2010 in this case) (LR 289-290). This approach is commonly used with different types of models (Ji et al., 2020; Xiao et al., 2021; Zhao et al, 2021; Qi et al., 2022), including CESM / CAMS (Banks et al., 2022) (LR 290-292). CESM / CAMS were also used to determine meteorological effects on AOD (Kang et al., 2019; de Leeuw et al., 2023) and in the current paper we followed up on De Leeuw et al. (2023) but with substantial differences (see our response to SC1), as described in the MS (LR 148-163). None of the publications mentioned above discusses the sensitivity to the resolution of the emission inventory or the nudging method and indeed, also in our study we did not include any sensitivity checks.

Your comment prompted us to consider the possibility of possible biases due to the assumption that all changes are due to variations in meteorological parameters. However, the evaluation of the sensitivity of the model results to the emission inventory resolution or nudging methods applied would be a major task going far beyond the objectives of the current study. In particular because the methods we used are similar to those published in other research papers. However, the questions you raised may have important consequences for the interpretation of our results and those from

similar research and may therefore have been addressed in earlier research. In response, we have searched the literature for such effects on aerosol simulations using CESM / CAM but did not find references specifically addressing these topics. However, several papers address nudging and resolution.

As regards nudging, Menut et al. (2024), using WRF Chem, showed that the use of nudging significantly improves the model performances. The importance of nudging for CAM5 was discussed by Zhang et al. (2014) who show the improvement of the top-of-atmosphere radiation budget and cloud ice amount. He et al. (2015) evaluated CESM focusing on the atmospheric component CAM5.1 and concluded that most meteorological and radiative variables are relatively well reproduced with normalized mean biases (NMBs) of 214.1 to 29.7% and 0.7–10.8%, respectively, and mentioned the good performance for liquid water path (LWP) and AOD. In the revised MS we have added "The importance of nudging was discussed in, e.g. Menut et al. (2024), Zhang et al. (2014) and He et al. (2015)." (LR 284-285)

As regards resolution, Bacmeister et al. (2014) demonstrated to some extent that increasing the resolution of the CAM5 model does not necessarily lead to significant improvement in the simulation of long-term variations in meteorological variables. Huang et al. (2016) assessed the recently developed variable-resolution option within the Community Earth System Model (VR-CESM) for long-term regional climate modeling of California, for meteorological variables. Glotfelty et al. (2017) evaluated CESM and concluded that ESM-NCSU provides a reasonable representation of the current atmosphere and that biases in chemical predictions are due to inaccurate emissions, mixing, deposition, and volatility of primary organic aerosol. In the revised MS we have added "This approach is commonly used with different types of models (Ji et al., 2020; Xiao et al., 2021; Zhao et al, 2021; Qi et al., 2022), including CESM / CAM (Banks et al., 2022; Kang et al., 2019; de Leeuw et al., 2023). Model resolution was addressed by, e.g., Bacmeister et al. (2014), Huang et al. (2016) and Glotfelty, et al. (2017). " (LR 290-294)

We also added a reference to model performance "Model performance on aerosol has been widely evaluated (Lamarque et al., 2012; Fang et al., 2020; Emmons et al., 2010; He et al., 2015)" (LR271-272)

References:

- He, J., Y. Zhang, T. Glotfelty, R. He, R. Bennartz, J. Rausch, and K. Sartelet (2015), Decadal simulation and comprehensive evaluation of CESM/ CAM5.1 with advanced chemistry, aerosol microphysics, and aerosolcloud interactions, J. Adv. Model. Earth Syst., 7, 110–141, doi:10.1002/2014MS000360.
- Menut, L., Bessagnet, B., Cholakian, A., Siour, G., Mailler, S., and Pennel, R.: What is the relative impact of nudging and online coupling on meteorological variables, pollutant concentrations and aerosol optical properties?, Geosci. Model Dev., 17, 3645–3665, https://doi.org/10.5194/gmd-17-3645-2024, 2024
- Zhang, K., Wan, H., Liu, X., Ghan, S. J., Kooperman, G. J., Ma, P.-L., Rasch, P. J., Neubauer, D., and Lohmann, U.: Technical Note: On the use of nudging for aerosol–climate model intercomparison studies, Atmos. Chem. Phys., 14, 8631–8645, https://doi.org/10.5194/acp-14-8631-2014, 2014.
- Ji, H., Shao, M. and Wang, Q. (2020). Contribution of Meteorological Conditions to Inter-annual Variations in Air Quality during the Past Decade in Eastern China. Aerosol Air Qual. Res. 20: 2249–2259. https://doi.org/10.4209/aaqr.2019.12.0624

- Xiao, Q., Zheng, Y., Geng, G., Chen, C., Huang, X., Che, H., Zhang, X., He, K., and Zhang, Q.: Separating emission and meteorological contributions to long-term PM$_{2.5}$ trends over eastern China during 2000–2018, Atmos. Chem. Phys., 21, 9475–9496, https://doi.org/10.5194/acp-21-9475-2021, 2021.
- Zhao, Y., Huang, Y., Xie, F., Huang, X., Yang, Y. (2021). The effect of recent controls on emissions and aerosol pollution at city scale: A case study for Nanjing, China, Atmospheric Environment, 246, 118080, ISSN 1352-2310, https://doi.org/10.1016/j.atmosenv.2020.118080
- Qi, L.; Zheng, H.; Ding, D.; Ye, D.; Wang, S. Effects of Meteorology Changes on Inter-Annual Variations of Aerosol Optical Depth and Surface PM2.5 in China—Implications for PM2.5 Remote Sensing. Remote Sens. 2022, 14, 2762. https://doi.org/10.3390/rs14122762
- Banks, A., Kooperman, G. J., & Xu, Y. (2022). Meteorological influences on anthropogenic PM2.5 in future climates: Species level analysis in the Community Earth System Model v2. Earth's Future, 10, e2021EF002298. https://doi.org/10.1029/2021EF002298
- Bacmeister, J.T., Wehner, M.F., Neale, R.B., Gettelman, A., Hannay, C., Lauritzen, P.H., Caron, J.M., Truesdale, J.E., (2014). Exploratory High-Resolution Climate Simulations using the Community Atmosphere Model (CAM). J. of Climate 27, 3073-3099. DOI: https://doi.org/10.1175/JCLI-D-13-00387.1
- Huang, X., A. M. Rhoades, P. A. Ullrich, and C. M. Zarzycki (2016), An evaluation of the variable-resolution CESM for modeling California's climate, J. Adv. Model. Earth Syst., 8, 345-369, doi:10.1002/2015MS000559.
- Glotfelty, T., He, J., Zhang, Y. (2017). Impact of future climate policy scenarios on air quality and aerosol-cloud interactions using an advanced version of CESM/CAM5: Part I. model evaluation for the current decadal simulations, Atmospheric Environment, Volume 152, 2017, Pages 222-239, ISSN 1352-2310, https://doi.org/10.1016/j.atmosenv.2016.12.035.:

**SC5:** No Integration of Large-Scale Climate Drivers: the paper mentions El Niño/La Niña (e.g., L760) only in the discussion, but these large-scale drivers are not analyzed or incorporated in any statistical way. Given the known influence of ENSO and the East Asian monsoon on AOD, this omission is a missed opportunity for deeper interpretation.

**Response to SC5**: Thank you for this comment. The influences of El Niño/La Niña and other large scale phenomena on the AOD variations will be discussed in a follow-up paper, to avoid that the current MS becomes too long. We have added this information to Section 4 (Discussion) (LR 641-642; LR 711-727) and Section 5 (Conclusion) (LR 850-851). The influences vary across different regions, depending on weather patterns, large range transport from source regions and geographical effects influencing transport. That is also the reason why different regions were selected across different climate zones as mentioned in Section 2 (L 181-184), with different effects as discussed in the following lines (L185-195) and shown in Fig. 2.

However, to meet concerns expressed by you and Referee#2, we have provided more information on meteorological effects on the AOD evolution in Section 4: "Yin et al. (2017) ascribed the occurrence of severe winter haze events in the North China Plain in 2014 to a weakened East Asian winter monsoon (EAWM) and anticyclonic circulation. Wang and He (2015) ascribed the North China / Severe Summer Drought in 2014 to a weakened East Asian summer monsoon (EASM). A weak EASM results in increased aerosol concentrations over northern China (Feng et al., 2016). Effects of El Niño– Southern Oscillation (ENSO) on air quality in southern China (i.e., south of the Yangtze River) were described by Wang et al. (2022): anticyclonic circulation during El Niño events weakens EASM

resulting in low AOD. Vice versa, cyclonic circulation during La Niña events strengthens EASM resulting in high AOD. This may explain the stronger enhancement of the AOD in the PRD, YRD and HNB than in the NCP." (LR 711-719) (Section 4.2). In addition, Section 4.3 describes an analysis where observations and simulations are used to explain differences in aerosol properties and AOD between the periods 2010-2016 and 2018-2024. (LR 728-776).

**SC6:** Policy Implications Lacking Synthesis: while the abstract references "emission reduction policy," the conclusions do not explicitly synthesize what the results mean for China's air quality regulation or international climate targets. Clarifying how meteorological dominance may affect future policy planning would improve the relevance.

**Response to SC6**: Thank you for this comment. We have restructured the Discussion in Section 4 and added "Section 4.1 Overall effects of emission reduction policy on aerosol properties" (LR 649-685). We further discuss "AOD reduction over different regions between 2010 and 2018 and influences of anomalous meteorological situations" (Section 4.2) and "AOD variations after 2018 over different regions: increasing importance of meteorological influences" (Section 4.3). Section 5 (Conclusions) summarizes the effectiveness of China's air quality regulations during the study period.

However, the meaning of these results for international climate targets and how meteorological dominance may affect future policy planning is beyond the scope of the current study. The proper evaluation of these effects would require a separate study including future projections using climate scenarios.

Technical Corrections

L248: Clarify whether the "2010 level" refers to monthly averages or annual means.

Thank you for this comment. This has been clarified "anthropogenic emissions were fixed using the monthly values from 2010, which were repeatedly applied to the corresponding months of each subsequent year" (LR 273-274)

L263: Fix punctuation — the sentence ends with a comma.

Thank you for noticing this typo. We have replaced the comma by a full stop.

L322: Explain the method of normalization and its justification; currently unclear.

Thank you for this comment. We have added the following footnote at the first occurrence of "normalization" (LR 161): " Both model and satellite AOD time series were divided by their respective values in July 2010, i.e. at the start of the normalized time series, each of the normalized time series has the value 1, as illustrated in Figs. 5, 7, 9, 11 and 13. If there are no meteorological effects on the AOD, the model data points in the time series are all 1; any deviation from 1 indicates meteorological influences on the AOD. Any deviation between the satellite and model data indicates anthropogenic influences on the AOD."

Figures: Add all missing axis labels; clarify units in captions.

Thank you for this comment. All figures have been checked and axis labels have been added, or corrected. Could you please clarify what you mean with "units in captions"? AOD and normalized AOD are unitless.

Table 1: Consider adding land area or population for context on regional significance.

Thank you for this comment. However, as mentioned on (L184-195), there are large variations in AOD, as well as geographical differences delineating population density and industrialization. There are large regional differences. In De Leeuw et al. (2023) we did add a population density map that showed the variation across each region. In the current study we did not correlate population density, land area and other indicators which may influence AOD and therefore decided to not include this information in Table 1.

---

## Author Comment (AC3)

**Response to RC2**: 'Comment on egusphere-2025-880', Anonymous Referee #2, 14 June 2025

Review of Manuscript egusphere-2025-880 entitled '**Evolution of aerosol optical depth over China in 2010-2024: increasing importance of meteorological influences**' by Cheng Fan, Gerrit de Leeuw, Xiaoxi Yan, Jiantao Dong, Hanqing Kang, Chengwei Fang, Zhengqiang Li, Ying Zhang

On behalf of all co-authors, we thank Referee #2 for the insightful and extensive comments which certainly contribute to the substantial improvement of the manuscript (MS). Below we respond to each of the general, major and specific comments which are copied below (in black). In addition to the numbered major and specific comments, we have numbered the general comments as GC1-GC5. After each comment we provide our response, in red, together with changes in the revised MS. Line numbers (indicated by L) mentioned by Referee #2 refer to the original MS as published in the ACP discussion Section and revisions are quoted with line numbers (indicated by LR) referring to the revised MS.

**GC1:** This manuscript investigates the evolution of AOD in China and the impacts of meteorological and anthropogenic effects across five regions from January 2010 to September 2024, by using MODIS/MAIAC C6.1 data and the CESM CAM5 model, incorporating MERRA-2 meteorological parameters and aerosol emission data as inputs. Processed MAIAC AOD data are used to provide observational insights, while simulated AOD data specifically reflect the influence of meteorological factors. The study focuses on five key regions, i.e., NCP, YRD, PRD, HNB, and SCB. The methodology builds upon a previous study by de Leeuw et al. (2023). The authors extend their AOD dataset from 2010-2021 in de Leeuw et al. (2023) to 2010-2024 in this study. The authors highlight the increasing significance of meteorological factors in AOD variation in recent years.

**Response to GC1**: Thank you for this summary. However, we would like to clarify that the AOD dataset from 2010-2021 in de Leeuw et al. (2023) was MAIAC C6. In this study, the 2010-2021 MAIAC C6 has been replaced with MAIAC C6.1 and then extended to 2024. This implies that the whole AOD data set is C6.1, hence it is a new data set and not merely an extension of the C6 data set used by de Leeuw et al. (2023). We have reformulated the text to "the C6 time series for 2010-2021 used by de Leeuw et al. (2023) was replaced with the recently released (6 July 2022) MODIS/MAIAC C6.1 data and extended with C6.1 data until September 2024." (LR 142-143). Hence a new data set has been used for the whole study period. MAIAC C6.1 is different from MAIAC C6 (L 139-143), (LR 144-147). We further refer to our Response to specific comment 8 and our Response to specific comment 11.

**GC2:** First of all, I must admit that reading the article is quite challenging due to issues with both logic and language, making me always struggling to grasp the main points of individual paragraphs, which suggests that the manuscript lacks clear organization. Before (required and recommended) re-submission, I strongly encourage the authors to seek professional assistance from a native speaker or a language editing service.

**Response to GC2**: Thank you for these comments. We have substantially revised the MS, not only following your specific comments but also keeping the above comment in mind, with the aim to improve the logic and organization. In particular, the discussion on Section 4 has been re-organized and sub-headings have been added to provide a clear structure. Furthermore, the manuscript has been carefully read and where necessary, unclear text has been re-formulated.

**GC3:** Many paragraphs in Sections 3 and 4 are overly wordy, particularly those discussing long-term AOD variations across different regions. Since this work is an extension of de Leeuw et al. (2023), with the primary addition being three more years of data (2022-2024), it is highly recommended that the authors avoid presenting exhaustive details on all the long-term, short-term, and even occasional

monthly anomaly of AOD trends. Such extensive reporting would bring a risk of obscuring the key messages of this study. Instead, the focus should be on the new insights that differ meaningfully from the previous work by de Leeuw et al. (2023).

**Response to GC3**: Thank you for these comments. Apparently. our formulation was not clear at this point, which caused a misunderstanding as is also indicated by your first comment GC1. We sincerely apologize for causing this misunderstanding which may also have influenced the rest of your review. We refer to our Response to GC1 for changes made in the revised MS to explain this better. Indeed, we applied methods similar to those of de Leeuw et al (2023), but we used different data and instead of the KZ(12,3) filter, which restricted the analysis to very long time periods, we used the CMA12 filter to remove seasonal and monthly variations, together with monthly data to pinpoint the months when events occurred that influence the AOD evolution (L135-150). This study highlights major differences of the increasing importance of meteorological influences to air pollution (AOD) compared with the previous study. Moreover, in Section 3 (Results) and Section 4 (Discussion, in particular Section 4.3) of this study, the focus is on the identification of the differences between meteorological and anthropogenic effects to support our conclusion of the increasing importance of meteorological effects (LR761-765). Without the detailed explanations in Section 3, it would be difficult for readers to discriminate between AOD changes due to anthropogenic effects and meteorological influences, and finally may fail to get the point of our conclusion. Therefore, we believe it is important to keep most of Sections 3 and 4. But we also have to admit that some places are not organized quite well (especially Section 4), so as you suggested, we reorganized Section 4 and made some modifications in Section 3 to avoid misunderstandings.

In addition, we modified the Section where we summarized the new findings in Section 4 (L712-723) and moved it to Section 5 (LR 840-851). Furthermore, as suggested by another Referee we added "Objectives" to Section 1 "The objectives of the current study are (1) to investigate the reasons for the flattening of the AOD reduction during 2017-2021, observed by De Leeuw et al. (2023); (2) to investigate what caused the anomalous AOD in the winter of 2014 over the YRD, HNB and PRD, but not over the NCP and SCB (De Leeuw et al., 2023; Fig. 7); (3) to use monthly mean AOD data to accurately identify the start and end of anomalous events, which are hidden in the low-pass filtered data used in De Leeuw et al. (2023); (4) to connect the occurrences of anomalous AOD to specific meteorological conditions and/or anthropogenic interferences; (5) to investigate whether changes in aerosol physicochemical characteristics, in response to emission reduction and climate change, results in different AOD patterns.

"Obviously, not all of these questions can be fully addressed in a single study. In the current paper we report and describe the observational data and provide comparisons with the CESM model data (with emissions fixed in 2010). Meteorological and anthropogenic effects on the AOD variations are discussed and possible influences of large scale meteorological effects (El Niño, La Niña, heat waves) and anthropogenic effects (policy measures and economic effects) are indicated. These effects will be discussed in more detail in a follow-up paper." (LR 181-186) and added text describing how we addressed these to Section 5: "The objectives stated in the Introduction were addressed throughout this paper. Data presented and discussed show that the flattening of the AOD reduction between 2017 and 2021 suggested by De Leeuw et al. (2023) was a consequence of the offset of AOD reduction by unfavorable meteorological effects (Sections 3.3.1 and 4.3) (Objective 1), as also observed during earlier periods. The anomalous AOD in the winter of 2014 over the YRD, HNB and PRD has been explained by large scale meteorological effects influencing AOD, in particular by ENSO and East Asian winter and summer monsoon (Section 4.2) (Objective 2). Relations between anomalous AOD and meteorological situations have been discussed in Sections 4.2, 4.3 and 4.4 (Objective 4). Changing AOD patterns have been reported and suggested to be due to changing

atmospheric composition in response to selective emission reduction policy (Section 4.1), together with the occurrence of anomalous meteorological situations (Section 4.3) (Objective 5). Monthly mean AOD data were used throughout the paper to identify the occurrence of specific events (Objective 3)." (LR 852-863).

**GC4:** Furthermore, the conclusions regarding the influences of meteorological factors on AOD variations across different regions are not clearly articulated. Even in the abstract part, the use of phrase like 'for instance' introduces fragmented examples without conveying coherent or generalizable patterns.

**Response to GC4**: Thank you for this comment. In the abstract, we have given two examples which are discussed in detail in the text (L 34-39). In view of the ACP abstract limitation to 250 words, a more elaborate account of the differences is not possible. We have followed your advice and removed "For instance". Furthermore, meteorological effects do not show generalizable patterns (apart from "weather conditions conducive of the development of haze (low wind speed, low ABLH, etc., or lower/higher AOD associated with El Niño/ La Niña). The influences of such conditions on the AOD variations will be discussed in a follow-up paper, to avoid that the current MS become too long. We have added this information to Section 1 (LR 185-186), Section 4 (LR 641-642) and Section 5 (LR 850-851). The influences vary across different regions, depending on weather patterns, large range transport from source regions and geographical effects influencing transport. That is also the reason why different regions were selected across different climate zones as mentioned in Section 2 (L 181-184), with different effects as discussed in the following lines (L185-195) and shown in Fig. 2.

**GC5:** To ensure the statistical results are persuasive and the manuscript is readable, substantial revisions are necessary before this manuscript can be reconsidered for publication.

**Response to GC5**: Thank you for these general comments. The MS has been substantially revised, as detailed in our responses to individual comments: general comments above and below to major and specific comments.

**Major comments**

1. It would be helpful to clarify define the terms "meteorological effect", "anthropogenic effect", "favorable meteorological effect", and "unfavorable meteorological effect", as their meanings are not always clear (such as in L132-134, L328-330).

**Response to major comment 1**: Meteorological effects and anthropogenic effects in the context of this manuscript (MS) are effects on AOD due to meteorological and anthropogenic factors, respectively. Meteorological processes affecting aerosols were briefly indicated in the Introduction (L 55-61), and referred to at various instances throughout the MS. Anthropogenic effects are explained in L 109-112. Furthermore, the text in L 150-154 explains that the model runs are made with emissions fixed but with actual meteorological conditions and therefore all AOD changes in the model results are due to meteorological effects. Anthropogenic effects are the difference between actual (observed) and modeled changes.

Favorable meteorological effects are commonly used in studies on air pollution to indicate that concentrations of pollutants decrease during certain meteorological conditions (such as high wind speed, wind direction from clean areas, increasing ABLH, etc.) and vice versa for unfavorable meteorological conditions. See, for instance, Xiao et al. (2021), https://doi.org/10.5194/acp-21-9475-2021 (in list of references). These terms were explained in L 132-134.

2. The introduction Section spans nearly four pages, which is lengthy and not necessary. It is recommended to condense the general background into one or two concise paragraphs, and then promptly introduce the motivation and objectives of this study.

**Response to major comment 2**: It is common that the introduction of a scientific paper reflects the background and current status of the work described in the paper as well as what the MS contributes to advance the knowledge on the subject. In this case, the Introduction provides general information on aerosols (L48-112), starting with a brief overview of what aerosols are, their generation, physical properties and meteorological factors that affect these (48-61). In turn, effects of aerosols on climate and air quality (AQ) are briefly described and different measures for aerosol concentrations (AOD and $PM_{2.5}$) are explained (L62-78; 79-91). This is relevant for the current study, because $PM_{2.5}$ and AOD are both affected by meteorological influences, but in different ways and readers may wonder why the results presented here differ from those for $PM_{2.5}$. Then we introduce satellite observations relevant for the current study (L92-106).

The next 2 pages (L113-174) describe the background of the current study, starting with the evolution of pollutant concentrations (113-119) and what influences these. This is briefly summarized based on the previous study by de Leeuw et al. (2023), which considered KZ(12,3) filtered (21 moths) AOD rather than the CMA12 (12 months) and monthly mean data considered in the current study (120-134). Then we introduce how we build on this work in the current study and the major differences between the two studies (L135-150) followed by a brief summary of the approach (L150-158) and what can be expected in this paper (159-166) which is outlined in L168-174.

This brief summary shows the organization and motivation for this study, as well as how it is different from the previous study.

3. Section 2 lacks some important details regarding data processing. For example, the method used to obtain the CMA12 monthly mean AOD should be explained. Additionally, sentences from L142-147 would be more appropriate in the methodology Section. For the use of the CESM model, more detailed information should be provided, such as how the model is run. Since the model and input data have different spatial and temporal resolutions, the data processing procedure should be described clearly. It is also suggested to provide a flowchart in Section 2.3 so as to clearly illustrate the modeling process.

**Response to major comment 3**: We appreciate the reviewer's suggestion. As the CESM model configuration and simulation procedures in this study follow established practices, and detailed descriptions of emission settings, meteorological forcing, spatial and temporal resolution, and data handling have been added in the revised text, we believe the modeling process is now sufficiently clear. Specifically, the meteorological input fields were taken from the MERRA-2 reanalysis dataset, which provides model-ready data for CESM simulations at four spatial resolutions (https://rda.ucar.edu/datasets/d313003/dataaccess/#). Among them, we selected the 1.9° × 2.5° resolution dataset, which matches the CESM model grid and ensures consistency without the need for additional spatial interpolation. The temporal resolution of the MERRA-2 data is 3 hours. Therefore, a flowchart may not be strictly necessary. The revision to the second paragraph of Section 2.3 is as follows:

"To isolate the effects of meteorology on AOD, anthropogenic emissions were fixed using the monthly values from 2010, which were repeatedly applied to the corresponding months of each subsequent year. Meteorological input fields, including horizontal winds, air temperature, surface pressure, land surface temperature, heat fluxes, and wind stresses, were nudged to the MERRA-2 (Modern Era Retrospective analysis for Research and Applications, Version 2) reanalysis dataset (Gelaro et al., 2017; Rienecker et

al., 2011) (see also https://rda.ucar.edu/datasets/d313003/; last access 4 March 2024), which provides data at a 3-hour temporal resolution. In this study, we used the MERRA-2 product available at 1.9° × 2.5° horizontal resolution, which matches the CESM model grid and avoids the need for spatial interpolation. Linear interpolation in time was applied between input steps to ensure continuity and avoid artificial jumps (Lamarque et al., 2012). CAM5 employs a sub-stepping algorithm (Lauritzen et al., 2011) and an atmospheric mass fixer (Rotman et al., 2004) to maintain consistency between nudged and prognostic fields. The importance of nudging was discussed in, e.g., Menut et al. (2024) and Zhang et al. (2014).

Natural emissions of dust and sea salt were calculated online in the model using the actual MERRA-2 meteorological conditions. Biomass burning emissions from the Global Fire Emissions Database version 2 (GFEDv2) (Randerson et al., 2006) were treated as anthropogenic (Yan et al., 2006; Wu et al., 2020) and fixed at the 2010 level. As a result, all variations in the simulated AOD can be attributed to changes in meteorological parameters and their influence on natural aerosol processes. This approach is commonly used with different types of models (Ji et al., 2020; Xiao et al., 2021; Zhao et al, 2021; Qi et al., 2022), including CESM / CAMS (Banks et al., 2022; Kang et al., 2019; de Leeuw et al., 2023). Model resolution was addressed by, e.g., Bacmeister et al. (2014), Huang et al. (2016) and Glotfelty, et al. (2017)." (LR 273-294).

The method used to obtain the CMA12 (centered moving average, L145) monthly mean AOD is a standard procedure which can easily be found on the internet or in text books. For clarity we have added the equation used as footnote at the first occurrence of CMA12 (LR 150)

4. Section 3.1 provides a brief overview of AOD values across different regions but does not describe the long-term characteristics. It is recommended to merge Sections 3.1 and 3.2 as well as to merge Figures 2 and 3.

**Response to major comment 4**: Thank you for this comment. However, this is how we organized the paper: first provide an overview of the data (which are monthly means) and indicate some extremes, and that they occur at different times. Fig. 2 shows that individual characteristics are hard to see and therefore we smoothed the data by using centered moving averages over 12 months (CMA12), which removes monthly and seasonal variations, and compare the results for the 5 regions in Figure 3. Merging Figures 2 and 3 is a good suggestion and we tried that. However, the result is not clear (different lines are difficult to distinguish) and therefore we decided to present these figures separately. In the next Section (3.3) we organized the data and time series (monthly mean and CMA12) by region and discuss the features for each region separately. By doing that in different Subsections with different titles, we have a clear organization. The results are discussed in Section 4.

5. Section 3.3 should be significantly shortened, particularly the description of AOD variation, to avoid presenting detailed short-term AOD variation patterns for every region. As mentioned before, the focus should shift toward highlighting new findings, especially those resulting from the additional data covering 2022-2024, which extend the study by de Leeuw et al. (2023). Moreover, this Section only presents the AOD variation patterns and the influence of meteorological effects without offering any in-depth analysis. Since all discussions are provided in Section 4, making the readers very difficult to follow. It is suggested that explanations and interpretations for every region be included in the corresponding Subsection of Section 3, while Section 4 should focus solely on comparative analysis between regions.

**Response to major comment 5**:

The first part of this comment seems to originate in part from the misunderstanding addressed in our responses to GC1 and GC3. With the new data in C6.1, the results are somewhat different from those presented in de Leeuw et al. (2023), also because de Leeuw et al. (2023) considered only long term effects, whereas the current study considers meteorological influences which often occur on shorter time scales (L151-155). Therefore, the time series 2010-2024 is discussed with a focus on new findings. For differences between C6 and C6.1, see our Response to specific comment 8.

As regards your suggestion "that explanations and interpretations for every region be included in the corresponding Subsection of Section 3, while Section 4 should focus solely on comparative analysis between regions", this is what has been done and should be clearer after we restructured Section 4. Section 3.3 presents the data and focuses on regional aspects, whereas Section 4 focuses on discussing the data and common aspects and processes, where differences occur between regions. To better explain this, text has been added in the revised version, as a preamble in Section 4: "In Section 3.3 the AOD time series were discussed for each of the five regions separately. Below, results common to two or more regions are discussed, for different periods of time with distinct features. Meteorological events influencing AOD variations are indicated but a detailed analysis is outside the scope of the current study and will be presented in a separate publication. In this study, both monthly mean and low-pass filtered monthly mean (CMA12) AOD data have been used. Monthly mean time series show more detail than the CMA12 time series as regards the start and end time of events affecting the evolution of the AOD. The monthly data points in CMA12 time series represent the average over 12 months around that data point and remove short term features. CMA12 time series clearly show tendencies which are difficult to determine in monthly mean time series due to the monthly and seasonal variations." (LR 639-648).

Section 3 is dedicated to the presentation of the data. In Section 3.3 we discuss the features for each region in different Subsections and compare the smoothed (CMA12) time series with monthly mean data (as suggested in your major comment 4), as well as meteorological and anthropogenic influences. As mentioned on L 146-147: "The CMA12 time series reveals tendencies and variations which were further investigated using monthly mean data." The reason for analyzing both CMA12 and monthly mean data is explained in L707-711: "It is noted that the anthropogenic effects, such as those during the 2022 minimum, were confirmed by the monthly mean time series which showed more detail than the CMA12 time series as regards the start and end time of the AOD variations. The CMA12 time series smear the effects over a year and thus dilute them. Nevertheless, tendencies become clear in such time series and are difficult to determine in monthly mean time series due to the monthly and seasonal variations." (moved to the pre-amble of Section 4, and changed to "In this study both monthly mean and low-pass filtered monthly mean (CMA12) AOD data have been used. Monthly mean time series show more detail than the CMA12 time series as regards the start and end time of events affecting the evolution of the AOD. The monthly data points in CMA12 time series represent the average over 12 months around that data point and remove short term features. CMA12 time series clearly show tendencies which are difficult to determine in monthly mean time series due to the monthly and seasonal variations." (LR 642-648)

We start Section 3.3. with Subsection 3.3.1 presenting and explaining data over the NCP and, because it is the first, here we provide detail on the observational and model data (2 pp text); the following Subsections, presenting data over the other four regions, are shorter.

The in-depth analysis is outside the scope of the current study, as explained in our response to GC4 where we referred to LR 185-186, LR 641-642 and LR 850-851. However, we have provided more information on meteorological effects on the AOD evolution in Section 4: "Yin et al. (2017) ascribed the occurrence of severe winter haze events in the North China Plain in 2014 to a weakened East

Asian winter monsoon (EAWM) and anticyclonic circulation. Wang and He (2015) ascribed the North China / Severe Summer Drought in 2014 to a weakened East Asian summer monsoon (EASM). A weak EASM results in increased aerosol concentrations over northern China (Feng et al., 2016). Effects of El Niño–Southern Oscillation (ENSO) on air quality in southern China (i.e., south of the Yangtze River) were described by Wang et al. (2022): anticyclonic circulation during El Niño events weakens EASM resulting in low AOD. Vice versa, cyclonic circulation during La Niña events strengthens EASM resulting in high AOD. This may explain the stronger enhancement of the AOD in the PRD, YRD and HNB than in the NCP." (LR 711-719) (Section 4.2). In addition, Section 4.3 describes an analysis where observations and simulations are used to explain differences in aerosol properties and AOD between the periods 2010-2016 and 2018-2024. (LR 746-760).

6.  The logical structure of Section 4 is unclear and often confusing. The discussions frequently jump between different regions, from monthly to annual variations, and between long-term and short-term variation trends. The discussion also lacks a consistent chronological order. Thus, the main conclusions of the study are difficult to discern. Please clarify the core findings more clearly. Consider organizing the Section chronologically or dividing it into clearly defined thematic paragraphs to present different points in a more structured way.

**Response to major comment 6**: The structure of Section 4 has been drastically changed: a pre-amble has been added and 4 Subsections have been created with titles indicating the content of each of them. Text has been moved, revised and new text has been added.

**Specific comments:**

1.  Abstract, there should be an overall description of AOD evolution for the five areas during the whole studying period of 2010-2024, rather than start from "2018-2021 (L31)" or "after 2016 (L36)". Besides, some statements are not rigorous and confusing, for instance, "the data show (L37)", "over some areas (L38)", please describe in particular. Rephrase the sentence in L31-32, which is confusing.

**Response to specific comment 1**: Thank you for this comment. The sentence on L31-32 has been rephrased as "The large increment of the AOD over the YRD during 2018 - 2021 shows that the influence of meteorological effects on the aerosol load increases as AOD declines" (LR 31-32).

The sentence on L37 has been modified to "a strong AOD minimum is observed in 2022" (LR 37) With these changes the word count is at the limit, so we invite the reader to discover in the MS which of "some areas " (L38) are affected.

As regards your comment "there should be an overall description of AOD evolution for the five areas during the whole studying period of 2010-2024": the abstract has a strict limitation to 250 words. We present new insights from the current study, with the most important first, rather than in a chronological order. This follows your general comment GC3 "the focus should be on the new insights that differ meaningfully from the previous work by de Leeuw et al. (2023)" but does not leave room for an overall description as indicated in your comment. This is also in line with your general comment GC3 "it is highly recommended that the authors avoid presenting exhaustive details on all the long-term, short-term, and even occasional monthly anomaly of AOD trends." However, when we reorganized Section 4, we have added Subsection 4.1 with the tile "Overall effects of emission reduction policy on aerosol properties" (LR 649)

2. L98-100, this sentence seems irrelevant here. In this study, only spaceborne measurements of AOD are applied (rather gas as mentioned here).

**Response to specific comment 2**: Thank you for this comment that allows us to explain why we include some sentences on trace gases. Trace gases such as $SO_2$, $NO_2$, $NH_3$, VOCs are aerosol precursors for secondary aerosol formation (L50). Hence, knowledge on the concentrations of these precursor gases is important to understand the evolution of aerosol concentrations and their chemical and physical properties. The change of precursor gas concentrations in response to emission reduction policy (L98-119) has resulted in changes of aerosol properties (L612-620), which is a factor that needs to be included. However, "The model simulations were made with fixed emissions and thus cannot reproduce this situation. " (L620-621). We have added " In future studies, changes in aerosol properties in response to reduced emissions of precursor gases and associated chemical processes should be taken into account." (LR 684-685)

3. L118, add a reference for ground-based measurements.

**Response to specific comment 3**: We have added the following references: Zheng et al. (2017), Zhang et al. (2019), Xiao et al. (2020), Zhang et al. (2019), Zhong et al. (2021), Geng et al. (2024) (see references listed at the end of this document)

4. L120, it seems no need to mention trace gases.

**Response to specific comment 4**: See response to Specific comment 2.

5. L132-133, provide some basic explanations for how meteorological effects can "enhance" and "reduce" AOD? At least give some examples, such as more precipitation may remove more aerosols (reduce), and more humidity atmosphere can promote hygroscopic growth (enhance)…

**Response to specific comment 5**: Thank you for this comment. How meteorological effects can "enhance" and "reduce" AOD is mentioned in L54-61, and L120. In addition, the basic explanations you mention were provided in de Leeuw et al. (2023). Because the Introduction is quite long (major comment 2), and to avoid duplication, the following sentence has been added in the first paragraph of the Introduction (L54-61): "Effects of changes in meteorological parameters on AOD were explained in de Leeuw et al. (2023) and this information was used by these authors to explain changes in AOD time series (their Section 3.6)." (LR 57-59).

6. L135, please first mention the time period of de Leeuw et al. (2023), i.e., 2010-2021, and then the 3-year extension 2022-2024

**Response to specific comment 6**: We have added "for the time period 2010-2021" to L135, which now reads "The current study extends the work presented in De Leeuw et al. (2023) for the time period 2010-2021 with almost 3 years by adding MAIAC AOD data from the end of 2021 to until September 2024" (LR 138-139). See also our Response to GC1, GC3 and major comment 5

7. L138, explain "whole time series"

**Response to specific comment 7**: Thank you for this comment. The sentence has been replaced with "the C6 time series for 2010-2021 used by de Leeuw et al. (2023) was replaced with the recently released (6 July 2022) MODIS/MAIAC C6.1 data and extended with C6.1 data until September 2024." (LR 142-143).

8. L142-143, "somewhat different from", do you respond to this question later in the manuscript?

**Response to specific comment 8**: Differences between C6 and C6.1 have been published in papers by Lyapustin and Wang (2022) , LPDAAC (2024), Ji et al. (2024), Huang et al. (2024). These differences are briefly described in Section 2.2. We have also addressed the differences between C6 and C6.1 with a focus on the 5 areas used in our study, and in more recent years (until 2024) than used by Li et al. (until 2014) and Huang et al. (until 2021) However, including the analysis of the differences between C6.1 and C6 would render the MS very long and result in the loss of focus. Therefore, we decided to remove the analysis of the differences and prepare a separate paper where we also include validation results using other data sets (SONET), for other years and other areas than Li et al. and Huang et al. We have added a reference to this work "(Fan et al., 2025; in preparation)" (LR 147). An extensive description of these differences is beyond the scope of the current paper, since we do not make specific comparisons with de Leeuw et al (2023), for reasons mentioned in L135-147) (see also our response to Major comment 2.

9. L165-167, this sentence looks like the conclusion of the study and thus should not be shown in the introduction part.

**Response to specific comment 9**: Thank you for this comment. This is an important finding which should be included in the introduction where we summarize the content of the MS.

10. L228, "with"->"from"

**Response to specific comment 10**: Thank you for this comment. We have followed your advice.  (LR 247).

11. L233-234, for the evaluation of the MAIAC AOD product, "good enough" is actually enough. Please provide quantitative results from these two references.

**Response to specific comment 11**: The validation by Ji et al. (2024) was made over bright surfaces, which are outside our five study areas, and includes data which were mostly collected before our study period. Hence these results have general relevance for the assessment of the accuracy of MAIAC C6.1 AOD data (and therefore this reference is included) but are not relevant for our study. The validation by Huang et al. (2024) includes data in our study areas and overlaps in time (until 2021) with most of our study period. We have included the statistical metrics provided by Huang et al. (2024) and the paragraph now reads: "MAIAC C6.1 has been validated over China by Ji et al. (2024) and Huang et al. (2024). Both studies report that the overall accuracy of the MAIAC AOD products over China is good. The validation by Ji et al. (2024) over bright surfaces, using publicly available reference data from AERONET and CARSNET until 2014, shows a significant underestimation and negative bias of the MAIAC C6.1 product, which however performs slightly better than DB and C6. The comparison with collocated AERONET AOD data, for the period from 2001 to 2021, by Huang et al. (2024) shows good consistency, with correlation coefficients (R) of 0.933/0.939, root mean square error (RMSE) of 0.152/0.146, bias of 0.005/0.015, mean absolute error (MAE) of 0.094/0.092, relative mean bias (RMB) of 1.221/1.301 and percentage of data points within expected error (EE) of 71.02/68.36. These statistical metrics refer to comparison at the overpass times of the Aqua (13:30 LT) and Terra (10:30 satellites, respectively (Huang et al., 2024, Fig. 2). The comparison shows a slight overestimation of C6.1 at low AOD (<0.5) and a small underestimation at higher AOD." (LR 252-263). We further refer to our Response to specific comment 8 for more detailed validation which we made specifically over the five study areas in this paper.

12. L242, how do you handle the different resolutions of the CESM model and input data? Maybe provide more descriptions.

**Response to specific comment 12**: Thank you for this comment. Please refer to the response to major comment 3.

13. L249, not "actual" but from MERRA-2 data

**Response to specific comment 13**: Thank you for the clarification. We agree that the term "actual" may be misleading in this context. As suggested, we have revised the sentence to explicitly state that the meteorological fields were nudged to the MERRA-2 reanalysis data, rather than using the term "actual". This sentence was changed to:

"To isolate the effects of meteorology on AOD, anthropogenic emissions were fixed using the monthly values from 2010, which were repeatedly applied to the corresponding months of each subsequent year. Meteorological input fields, including horizontal winds, air temperature, surface pressure, land surface temperature, heat fluxes, and wind stresses, were nudged to the MERRA-2 (Modern Era Retrospective analysis for Research and Applications, Version 2) reanalysis dataset (Gelaro et al., 2017; Rienecker et al., 2011) (see also https://rda.ucar.edu/datasets/d313003/; last access 4 March 2024), which provides data at a 3-hour temporal resolution. n this study, we used the MERRA-2 product available at 1.9° × 2.5° horizontal resolution, which matches the CESM model grid and avoids the need for spatial interpolation. Linear interpolation in time was applied between input steps to ensure continuity and avoid artificial jumps (Lamarque et al., 2012). CAM5 employs a sub-stepping algorithm (Lauritzen et al., 2011) and an atmospheric mass fixer (Rotman et al., 2004) to maintain consistency between nudged and prognostic fields. The importance of nudging was discussed in, e.g., Menut et al. (2024) and Zhang et al. (2014).

Natural emissions of dust and sea salt were calculated online in the model using the actual MERRA-2 meteorological conditions. Biomass burning emissions from the Global Fire Emissions Database version 2 (GFEDv2) (Randerson et al., 2006) were treated as anthropogenic (Yan et al., 2006; Wu et al., 2020) and fixed at the 2010 level. As a result, all variations in the simulated AOD can be attributed to changes in meteorological parameters and their influence on natural aerosol processes. This approach is commonly used with different types of models (Ji et al., 2020; Xiao et al., 2021; Zhao et al, 2021; Qi et al., 2022), including CESM / CAMS (Banks et al., 2022; Kang et al., 2019; de Leeuw et al., 2023). Model resolution was addressed by, e.g., Bacmeister et al. (2014), Huang et al. (2016) and Glotfelty, et al. (2017)." (LR 273-294)

14. L262-263, how do you obtain this conclusion? Please add the necessary references.

**Response to specific comment 14**: Thank you for this comment. In support of this conclusion, please consider the figure below, referred to as Figure R1. Figure R1 shows the CESM-simulated PM10 column concentrations (μg/m²) for the year 2020, both with (Figure R1a) and without (Figure R1b) the contribution from desert dust. We also present the corresponding AOD values with (Figure R1c) and without (Figure R1d) desert dust included. These results clearly show that the inclusion of desert dust leads to PM10 column concentrations and AOD values that are several times higher than those without dust, particularly over arid regions such as the Taklamakan and Gobi deserts.

In addition, previous studies have also reported that CESM (CAM5) tends to overestimate dust-related aerosol properties. For example, Wu et al. (2019) noted that "in general, CAM5 overestimates dust extinction over the Taklamakan and Gobi deserts in March-April-May (MAM), June-July-August (JJA), and September-October-November (SON) but underestimates dust extinction in December-

January-February (DJF) compared with observations." We have added this reference in the revised text to further justify our approach (LR 298).

[Figure]

Figure R1. The CESM-simulated PM10 column concentrations (µg/m²) (a, b) and AOD (c, d) for the year 2020, both with (a, c) and without (b, d) the contribution from desert dust.

Wu, M., Liu, X., Yang, K., Luo, T., Wang, Z., Wu, C., Zhang, K., Yu, H., and Darmenov, A.: Modeling dust in East Asia by CESM and sources of biases. Journal of Geophysical Research: Atmospheres, 124, 8043–8064. https://doi.org/10.1029/2019JD030799, 2019.

15. L248, "The CESM model simulations were made with anthropogenic emissions fixed at the 2010 level", please specify "the 2010 level". Is it the annual mean emission? Or consider using the monthly emission in 2010 to represent the values for the corresponding months of each subsequent year.

**Response to specific comment 15**: Thank you for this valuable comment. We agree that clarification is necessary. In our study, anthropogenic emissions were fixed using the monthly emissions from the year 2010, and these monthly values were repeatedly applied to the corresponding months in each subsequent simulation year. We have revised the sentence accordingly: "To isolate the effects of meteorology on AOD, anthropogenic emissions were fixed using the monthly values from 2010, which were repeatedly applied to the corresponding months of each subsequent year." (LR 273-274)

16. L262-263, "the CESM estimates for desert dust are too high and therefore contributions from desert dust were not included in the AOD calculations", does this lead to an overestimation or underestimate of anthropogenic contributions?

**Response to specific comment 16**: Thank you for your insightful question. It is indeed difficult to definitively determine whether excluding desert dust leads to an overestimation or underestimation of anthropogenic contributions. This largely depends on whether meteorological conditions in the studied period were more or less favorable for dust emission and transport.

If simulated desert dust concentrations increased over the years while observed AOD significantly declined, the relative contribution of anthropogenic emissions to the AOD decrease would appear larger. Conversely, if simulated dust also decreased, the anthropogenic contribution in the observed AOD reduction would be smaller.

Including desert dust in the calculations would make the estimated meteorological contribution almost entirely dependent on changes in dust loading, since simulated dust AOD is several times higher than the total AOD when dust is excluded. This would obscure the role of meteorological factors, such as changes in transport, diffusion, and deposition conditions, in affecting anthropogenic aerosol levels, especially in key regions like the North China Plain (NCP) and the Yangtze River Delta (YRD).

Therefore, to better isolate the impact of meteorology on AOD, we excluded desert dust in this study.

17. L263, the comma should be modified to a full stop.

**Response to specific comment 17**: Thank you for noticing this typo. We have replaced the comma by a full stop (LR 319).

18. L288, "deep minimum"->"plunge"

**Response to specific comment 18**: Thank you for this comment. We have replaced "a deep minimum" with "a short period with very low values" (LR 323).

19. L298, what kind of 'test'?

**Response to specific comment 19**: Tests "in which the June peak values were set to the local average of 0.5". We have rephrased the text on L 297-298 ("monthly mean AOD peaks in June (Fig. 2), but these peaks did not have a determining role as evidenced by tests in which the June peak values were set to the local average of 0.5.") with" monthly mean AOD peaks in June (Fig. 2), but these peaks did not have a large effect on the variation of the CMA12 AOD in 2014 (Figure 3). Replacement of the monthly mean peak values in June (1.07 and 1.05, respectively), with the local mean value of 0.5 resulted in lower values of the CMA12 but did not substantially change the shapes of the CMA12 time series during the 12 months affected." (LR 333-335).

20. L308, what is the meaning of "less effective" and "not effective"

**Response to specific comment 20**: We have replaced "less effective over the NCP and not effective over the SCB." With "while over the NCP the AOD increase was smaller and over the SCB there was no increase." (LR 345-346).

21. L322, how observational and simulated AOD be normalized?

**Response to specific comment 21**: This was explained in de Leeuw et al. (2023) and we have added ", as explained in de Leeuw et al., 2023)" At the first occurrence of "normalized" (LR 161) a footnote has been added with the following text "Both model and satellite AOD time series were divided by their respective values in 2010, i.e. at the start of the normalized time series, in this case in July 2010, each of the normalized time series has the value 1, as illustrated in Figs. 5, 7, 9, 11 and 13. If there are no meteorological effects on the AOD, the model data points in the time series are all 1; any deviation

from 1 indicates meteorological influences on the AOD. Any deviation between the satellite and model data indicates anthropogenic influences on the AOD.".

22. L345-349, reorganized these two sentences, which read a bit strange now.

**Response to specific comment 22:** Thank you for this comment. We have replaced these sentences with "During the first years, the simulated AOD is substantially lower than the observations, except in November/December 2010 and 2012-2015, when simulated and observed AOD are in good agreement. The discrepancies during the spring and summer are attributed to the omission of the effect of desert dust in the AOD calculations (Section 2.3), while also anomalous meteorological conditions may have influenced the aerosol properties in the NCP (Fang et al., 2020)" (LR 382-386).

23. L366, "a period of about 6 months", not clear.

**Response to specific comment 23**: We have replaced text on L393-394 with "descent into September and during the next 6 months the AOD remained relatively high" (LR 402-403).

24. Figures 4, 6, 8, 10, and 12 have no y-axis titles.

**Response to specific comment 24**: Thank you for this comment. We have added titles to the y-axes in these figures.

25. Figures 5, 7, 9, 11, and 13, the y-axis titles "AOD" is incorrect. Please refer to de Leeuw et al. (2023) (Figure 6 therein)

**Response to specific comment 25:** Thank you for this comment. We have changed both the primary and secondary y-axes in these figures.

26. L393-394, this sentence is not clear enough. What do you mean "increase"? favorable or unfavorable?

**Response to specific comment 26:** We have added "unfavorable" before meteorological effects and the sentence now reads "the unfavorable meteorological effects continued to increase until June" (LR 430).

27. L413, 423, two "regular AOD pattern" has different meanings.

**Response to specific comment 27**: Both on L413 and L423 "regular AOD pattern" is followed by some words explaining what we mean with that.

28. L417-418, explain the results in 2014 and 2017 mentioned here.

**Response to specific comment 28:** Here we guide the reader through what we observe in the simulated results: we are not explaining the details.

29. Figure 6, two peaks in June of 2012 and 2014 can be related to the emissions from local agricultural straw burning. A lot of related references can be found for these cases.

**Response to specific comment 29**: Thank you for this comment. However, in view of our Response to specific comment 28, we have added this information in the discussion "The high AOD peaks observed over northern, eastern and central China (Section 3.1) occurred mostly during the beginning of the study period, in June, which are attributable to emissions from agricultural straw burning Liu et al. (2020). The intensive implementation of the ban on open crop straw burning between 2013 and 2018 resulted in a declining trend of PM2.5 emissions in eastern and central China (Huang et al., 2021) which may explain why such high AOD maxima are not observed in June during

later years. During these later years, elevated AOD peaks are observed over the PRD in the spring which are attributed to the transport of biomass burning plumes from Indochina during specific weather patterns (Xue et al., 2025)." (LR 663-670).

30. L473, please explain what kind of meteorological influence (be specific).

**Response to specific comment 30**: Thank you for this comment. However, see our Response to specific comment 28 and the last part of our Response to specific comment 29, where we have added a reference to Xue et al., 2025).

31. L478-479, it seems that this sentence can be removed

**Response to specific comment 31**: Thank you for this comment. The data in Fig. 8 show that the simulated AOD was always substantially higher than the observed AOD. This suggests that the emissions were always high, i.e. the initial estimate for March 2010, which was used for each March in subsequent years, were high. Therefore, we added this sentence, but we agree that this was not clear. Therefore, we added ", suggesting that the initial anthropogenic emission estimates were high" (LR 516). Together with the clarification that "anthropogenic emissions were fixed using the monthly values from 2010, which were repeatedly applied to the corresponding months of each subsequent year." (LR 273-274), this provides a possible explanation for the repeatedly higher simulated AOD.

32. L533, you define different 'regular pattern' for each region. Please clarify this in Sections 3.3.1-3.3.5

**Response to specific comment 32**: Thank you for this comment. The sentence on L533 reads "In contrast to the model simulations, there is no regular pattern in the monthly mean MAIAC observational AOD data." This sentence refers to the text just above, where we discussed the model simulations, which starts with " The time series of the monthly mean model-simulated AOD over the HNB in Fig. 10 show a regular pattern with distinct peaks in March and minima in the summer centered around July. (L525-526). From this context it is clear what we mean with "regular pattern". See also our Response to specific comment 27, which referred to a similar comment on Sect. 3.3.2.

33. L536, "not regular as…", please provide some explanations.

**Response to specific comment 33**: see our response to specific comment 32: the explanation is in the text above.

34. L550-553, no explanations for the observed results.

**Response to specific comment 34**: See our Response to specific comment 28.

35. L571-581, when comparing the MAIAC AOD and CESM-simulated AOD, please provide some explanations for the differences.

**Response to specific comment 35**: See our Response to specific comment 28.

36. L608-612, provide some explanations for the regional differences

**Response to specific comment 36**: See our Responses to major comment 6 and 5, in particular "detailed analysis is outside the scope of the current study and will be presented in a separate publication. " However, in the modified Section 4, we have added many explanations which will be further elaborated in the mentioned separate publication.

37. Line 760-762, "The high AOD in 2014 has been suggested to be due to anomalous circulation associated with El Niño / La Niña and the strengths of the East Asian summer and winter

monsoon effects", however, there are no descriptions of these El Niño / La Niña effects in the previous analysis.

**Response to specific comment 37**: Thank you for this comment. We have re-structured Section 4 and in Section 4.2 (AOD reduction over different regions between 2010 and 2018 and influences of anomalous meteorological situations) we have added the following paragraph "Yin et al. (2017) ascribed the occurrence of severe winter haze events in the North China Plain in 2014 to a weakened East Asian winter monsoon (EAWM) and anticyclonic circulation. Wang and He (2015) ascribed the North China / Severe Summer Drought in 2014 to a weakened East Asian summer monsoon (EASM). A weak EASM results in increased aerosol concentrations over northern China (Feng et al., 2016). Effects of El Niño–Southern Oscillation (ENSO) on air quality in southern China (i.e., south of the Yangtze River) were described by Wang et al. (2022): anticyclonic circulation during El Niño events weakens EASM resulting in low AOD. Vice versa, cyclonic circulation during La Niña events strengthens EASM resulting in high AOD. This may explain the stronger enhancement of the AOD in the PRD, YRD and HNB than in the NCP." (LR 711-719).

Such influences have also been mentioned in parts of Section 4.3 (AOD variations after 2018 over different regions: increasing importance of meteorological influences) on LR 728-776.

**Citation**: https://doi.org/10.5194/egusphere-2025-880-RC2

**References:**

Yixuan Zheng, Tao Xue, Qiang Zhang, Guannan Geng, Dan Tong, Xin Li and Kebin He. (2017). Air quality improvements and health benefits from China's clean air action since 2013 . *Environ. Res. Lett.* 12 114020DOI 10.1088/1748-9326/aa8a32.

Zhang, Q., et al. 2019. Drivers of improved PM2.5 air quality in China from 2013 to 2017. PNAS | December 3, 2019 | vol. 116 | no. 49 | 24463–24469. www.pnas.org/cgi/doi/10.1073/pnas.1907956116.

Qingyang Xiao, Guannan Geng, Fengchao Liang, Xin Wang, Zhuo Lv, Yu Lei, Xiaomeng Huang, Qiang Zhang, Yang Liu, Kebin He. Changes in spatial patterns of PM2.5 pollution in China 2000–2018: Impact of clean air policies, Environment International, Volume 141, 2020, 105776, ISSN 0160-4120, https://doi.org/10.1016/j.envint.2020.105776.

Geng-etal-2024-Efficacy of China's clean air actions to tackle PM2.5 pollution between 2013 and 2020. Nature Geoscience | Volume 17 | October 2024 | 987–994. Nature Geoscience | Volume 17 | October 2024 | 987–994 988Article https://doi.org/10.1038/s41561-024-01540-z.

Zhong-etal-2021-PM2.5 reductions in Chinese cities from 2013 to 2019 remain significant despite the inflating effects of meteorological conditions. One Earth 4, 448–458, March 19, 2021 ª 2021 Elsevier Inc., https://doi.org/10.1016/j.oneear.2021.02.003

Zhang, X., Xu, X., Ding, Y., Liu, Y., Zhang, H., Wang, Y., Zhong, J., 2019. The impact of meteorological changes from 2013 to 2017 on PM2.5 mass reduction in key regions in China. Sci. China Earth Sci. 62, 1885–1902. https://doi.org/10.1007/s11430-019-9343-3.

Lauritzen, P. H., Ullrich, P. A., & Nair, R. D.: Atmospheric transport schemes: Desirable properties and a semi-Lagrangian view on finite-volume discretizations. In P. H. Lauritzen (Ed.), Numerical Techniques

for Global Atmospheric Models (pp. 185–251). Berlin: Springer. https://doi.org/10.1007/978-3-642-11640-7_8. 2011.

Rotman, D. A., Atherton, C. S., Bergmann, D. J., Cameron-Smith, P. J., Chuang, C. C., Connell, P. S., Dignon, J. E., Franz, A., Grant, K. E., Kinnison, D. E., Molenkamp, C. R., Proctor, D. D., and Tannahill, J. R.: IMPACT, the LLNL 3-D global atmospheric chemical transport model for the combined troposphere and stratosphere: Model description and analysis of ozone and other trace gases. Journal of Geophysical Research, 109, D04303, https://doi.org/10.1029/2002JD003155, 2004.

Wu, M., Liu, X., Yang, K., Luo, T., Wang, Z., Wu, C., Zhang, K., Yu, H., and Darmenov, A.: Modeling dust in East Asia by CESM and sources of biases. Journal of Geophysical Research: Atmospheres, 124, 8043–8064. https://doi.org/10.1029/2019JD030799, 2019.

Tong Liu, Guojun He, Alexis Kai Hon Lau, Statistical evidence on the impact of agricultural straw burning on urban air quality in China, Science of The Total Environment, Volume 711, 2020, 134633, ISSN 0048-9697, https://doi.org/10.1016/j.scitotenv.2019.134633.

Ling Huang, Yonghui Zhu, Qian Wang, Ansheng Zhu, Ziyi Liu, Yangjun Wang, David T. Allen, Li Li, Assessment of the effects of straw burning bans in China: Emissions, air quality, and health impacts, Science of The Total Environment, Volume 789, 2021, 147935, ISSN 0048-9697, https://doi.org/10.1016/j.scitotenv.2021.147935.

Xue, L., Ding, K., Huang, X., Zhu, A., Lou, S., Wang, Z., et al. (2025). Biomass burning plumes from Indochina toward southern China: Predominant synoptic weather processes and interactions. *Journal of Geophysical Research: Atmospheres*, 130, e2024JD041813. https://doi.org/10.1029/2024JD041813

Feng, J., Zhu, J., Li, Y. (2016). Influences of El Niño on aerosol concentrations over eastern China. Atmos. Sci. Let. 17: 422–430. https://doi.org/10.1002/asl.674.

Wang-etal-2022-Evaluation of the influence of El Niño–Southern Oscillation on air quality in southern China from long-term historical observations-2095-2201-2022-2-26

Zhang, C., Luo, J.-J., & Li, S. (2019). Impacts of tropical Indian and Atlantic Ocean warming on the occurrence of the 2017/2018 La Niña. Geophysical Research Letters, 46, 3435–3445. https://doi.org/10.1029/2019GL082280

Iwakiri, T., Imada, Y., Takaya, Y., Kataoka, T., Tatebe, H., & Watanabe, M. (2023). Triple-dip La Niña in 2020–23: North Pacific atmosphere drives 2nd year La Niña. Geophysical Research Letters, 50, e2023GL105763. https://doi. org/10.1029/2023GL105763

Tseng et al. (2024). https://doi.org/10.1016/j.atmosres.2024.107772)

---

## Author Response (AR2)

**Author's Response to 2nd Review of Anonymous Referee #2 (Report #1)**

Second review of Manuscript egusphere-2025-880 entitled 'Evolution of aerosol optical depth over China in 2010-2024: increasing importance of meteorological influences' by Cheng Fan, Gerrit de Leeuw, Xiaoxi Yan, Jiantao Dong, Hanqing Kang, Chengwei Fang, Zhengqiang Li, Ying Zhang

On behalf of all co-authors, we thank Referee #2 for providing a second review of our Manuscript egusphere-2025-880 and the insightful comments which contribute to further improvement. Below we respond to each of the comments which are copied below (in black). After each comment we provide our response, in red, together with changes in the revised MS. Line numbers (indicated by L) refer to t the revised MS.

The reviewer acknowledge that the authors clarify that they replaced the MAIAC C6 dataset used in de Leeuw et al. (2023) with MAIAC C6.1 for this study and extended the study period to include data after 2021. In the revised manuscript, they emphasize the differences compared to de Leeuw et al. (2023). Based on my evaluation, most of my concerns have been addressed through clarifications or modifications. However, there are a few remaining issues that still need consideration.

Thank you for this kind comment and indicating remaining issues.

In the conclusion, it is commendable that the authors divided the study period into two parts, i.e., before and after 2018. Consequently, there is potential to reorganize the abstract to better highlight the main findings of this study. While acknowledging the diverse meteorological impacts on AOD variations across five study areas, the abstract only mentions results from three of them. Quantitatively comparing meteorological influences on AOD variation before and after 2018, as demonstrated in Lines 128-133, would enhance clarity. This approach aligns with the manuscript's focus on the increasing importance of meteorological influences, as highlighted in both the manuscript's title and Section 4.3.

Thank you for this comment. Lines 128-133 summarize results from de Leeuw et al. (2023) using a different data set (C6 instead of C6.1 used in the current paper) and different methods (KZ(12,3) instead of CMA12 and monthly means) to evaluate the total AOD reduction between 2010 and 2020 and the contribution of meteorological effects to the total reduction. The objectives of the current study (L 174- 181) are aimed at investigating the reasons for deviations from the long term trends in de Leeuw et al. (2023). Therefore, we did not do the analysis of meteorological and anthropogenic contributions. However, we did mention the overall reduction of the AOD between the beginning (2010) and the end (2024) of the time series studied (L654-657). In this revised version we have disentangled the meteorological and anthropogenic contributions for the whole period and those before and after 2018, following methods described in de Leeuw et al. (2023). The results are presented in Table 2 and described in the text in Section 4.1 (L600-642):

"Satellite measurements of AOD over China show that emission reduction policy has been successful in reducing the aerosol concentrations between 2010 and 2018, with an additional but smaller reduction toward the end of the study period, in 2024. Over the NCP, the AOD in 2024 had been reduced to 68% of its value in 2010, over the YRD to 62%, over the PRD to 70%, over the HNB to 55% and over the SCB to 57% (CMA12 values). These reductions are larger than reported by De Leeuw et al. (2023) for the period July 2011-February 2020. The current study covers the longer period, with the larger reductions indicating that AOD was further reduced after February 2020. However, it should be kept in mind that, in the current study, a new data set and different methods were used, which may have influenced the results.

The data further show that the AOD differences between the five regions have become smaller. In 2010 the AOD over the five regions ranged from 0.40 (PRD) to 0.53 (YRD and HNB), while in 2024 the AOD over the five regions ranged from 0.29 to 0.33 (see Figure 3). However, a closer look shows that the AOD did not vary monotonously and substantial variations occurred, as revealed after low-pass filtering (CMA12). The AOD not only varied between the five different regions, but the AOD variations within each region occurred at different times as illustrated in Fig. 3. Clearly, not only

emission reduction policy and other anthropogenic factors (economic development, urbanization, etc.) influenced AOD but also meteorological factors. This is further illustrated by the analysis of anthropogenic and meteorological contributions to the AOD (cf. De Leeuw et al., 2023; Section 2.4.2) applied to the C6.1 data set. Because CESM data (available until July 2023) are used, the analysis was made for the period from January 2010 to July 2023, as well as the periods before and after 2018. Due to the CMA12 filtering, the period was reduced to July 2010 to January 2023. The results in Table 2 show that, over the whole period, meteorological contributions vary between 12% (NCP) and 33% (PRD), whereas over shorter periods they are overall larger, between 14% (SCB) and 31% (PRD) for the period until 2018 (Period 1) and 14% (YRD) to 52% (NCP) for the period after 2018 (Period 2). The data further show that during Period 2 the meteorological contributions were substantially larger than during Period 1: over the NCP (52% vs 16%), the PRD (43% vs 31%) and the SCB (38% vs 14%), whereas they are smaller over the YRD (14% vs 28%) and the HNB (21% vs 25%). In view of the large AOD increase between mid-2018 and January 2021 over the YRD and HNB (Fig. 3) the smaller meteorological contributions in Period 2 than in Period 1 may be surprising, However, the data in Fig. 3 also show the much higher AOD over these areas during extended periods in 2011 and 2014 with clear meteorological influences as shown in Fig. 7 for the YRD and Fig. 11 for the HNB, which may have resulted in relatively large meteorological contributions during Period 1.. When we isolate the period mid-2018 and January 2021, we find that the meteorological contributions were 33% over the YRD and 32% over the HNB.

The data in Table 2 show that the overall AOD reduction during the study period is mainly due to anthropogenic effects, most likely emission reduction policy. However, meteorological effects are substantial and their importance seems to increase as AOD becomes smaller. Their magnitude depends on the period analyzed and is connected with certain meteorological conditions. This will be further discussed in Sections 4.2 (before 2018) and 4.3 (after 2018).

**Table 2. Anthropogenic and meteorological contributions to the AOD variation over each of the 5 study areas for the whole study period and periods before and after 2018.***

| Period | | 7/2010-1/2023 | | 7/2010-6/2018 | | 7/2019-1/2023 | |
|---|---|---|---|---|---|---|---|
| Type of contribution (%) | Total reduction 1/2010-9/2024 | Anthrop. | Meteor | Anthrop. | Meteor | Anthrop. | Meteor |
| NCP | 68 | 88 | 12 | 84 | 16 | 48 | 52 |
| YRD | 62 | 83 | 17 | 72 | 28 | 86 | 14 |
| PRD | 70 | 67 | 33 | 69 | 31 | 57 | 43 |
| HNB | 55 | 84 | 16 | 75 | 25 | 79 | 21 |
| SCB | 57 | 90 | 10 | 86 | 14 | 62 | 38 |

In addition to the differences between the periods before and after 2018, the monthly mean MAIAC and"

As regards the abstract, we note that the allowed word count is 250, which does not leave enough space to summarize many details. To follow your suggestion, we had to delete much of the original abstract and re-organize the text. We have now included results from the above analysis for the total study period and the period before and after 2018, for all 5 regions, as per your suggestion. The abstract now reads (L26-41):

"Time series of MODIS/MAIAC C6.1 aerosol optical depth (AOD) and model-simulated AOD were used to determine contributions of meteorological and anthropogenic effects to spatiotemporal AOD variations over five representative areas in China, during the period January 2010 - September 2024. The time series confirm the effective reduction of the AOD between 2010 and 2018, with an additional but smaller reduction thereafter. The overall AOD reduction is mainly attributable to emission reduction policy, but with substantial meteorological effects. The total reduction and meteorological contributions during the whole study period, and the meteorological contributions before and after

2018 over the five regions were for NCP (68, 12, 16, 52) (all in %), YRD (62, 17, 28, 14), PRD (70, 33, 31, 43), HNB (55, 16, 25, 21), SCB (57, 10, 14, 38). Meteorological effects for each of these periods and each region are discussed in detail. As an example, the above data show that the meteorological effects over the YRD and HNB after 2018 are smaller than before 2018 which can be explained by the occurrence of strong effects in the earlier period and the choice of the period over which effects were calculated. Monthly mean AOD patterns were distinctly different before and after 2016, suggesting that aerosol properties changed in response to emission reduction policy. In summary, this study highlights the complex interplay between meteorological and anthropogenic factors in shaping AOD variations across China and demonstrates the increasing significance of meteorological conditions in modulating China's AOD".

Regarding the introduction section, I suggest the authors consider simplifying Lines 1-93. A lengthy exposition may not be necessary, starting instead with a brief introduction to aerosols and emphasizing the significance of AOD in assessing the atmospheric environment. Given the manuscript's current length, streamlining this section could enhance readability. In authors' responses they insisted on retaining this content; thus, I would leave the final decision to the editor's discretion.

Thank you for this comment. Indeed, we did insist retaining the Introduction as it was. However, in this second revision we decided to follow your advice. We have re-organized the Introduction, in particular removed much of the more general aerosol descriptions, PM2.5 and trace gases. We thus shortened the text by 1.5-2 pp. We now start directly with AOD. Lines 48-114 have been replaced with (L46-64): "Satellite observations of aerosol optical depth (AOD) provide information on the spatiotemporal variation of aerosols in the atmosphere on local, regional and global scales with daily global coverage. Satellite data have been used to retrieve AOD since 50 years and long time series are available from individual sensors such as the MODerate resolution Imaging Spectroradiometer (MODIS) and combinations of sensors (Sogacheva et al., 2020). The use of satellites to monitor the evolution of AOD over China has been demonstrated by, e.g., Xu et al. (2015), Kang et al. (2016), Zhang et al. (2017a), Zhao et al. (2017), Proestakis et al. (2018), De Leeuw et al. (2018; 2022; 2023), Sogacheva et al. (2018a; 2018b). Time series of aerosols provide information on the evolution of their atmospheric concentrations which are influenced by anthropogenic and natural emissions, transformations in the atmosphere and removal processes. Anthropogenic emissions include those due to, e.g., industrialization, urbanization, traffic, domestic activities and associated increase in energy production, transportation, agricultural activities, land use, etc. Emissions, and thus concentrations, are reduced by the implementation of policies aimed at the reduction of air pollution and its adverse effects. Effects of changes in meteorological parameters on AOD and associated effects on AOD time series were explained in De Leeuw et al. (2023) (their Section 3.6). Meteorological effects on AOD can be determined using model simulations, which in turn can be used together with observations to determine anthropogenic effects (Kang et al., 2019; De Leeuw et al., 2023). These methods, explained in more detail in Sections 2 and 3, are used in the current study on the analysis of AOD time series over China." and continues at L116 in the first revision (L97 in the current 2nd revision: "Early in the 21st century, aerosol and trace gas concentrations over China"

Regarding Lines 448-449, the mention of Shanghai in this context, related to the analysis of the North China Plain (NCP), appears inappropriate.

Thank you for this comment. However, this seems to be a misunderstanding. In this sentence we cite an article by Liu et al. (2024), the title of which is "Assessment of national economic repercussions from Shanghai's COVID-19 lockdown". We have changed the sentence to (L392-393): "Liu et al. (2024b) reported that Shanghai's lockdown in 2022 resulted in a stagnating economy in parts of China"

We made a similar change on line 451.

Regarding the PRD, where long-range transported smoke aerosols from South Asia are common, it is essential to consider how such non-local aerosol emissions may influence the conclusions drawn in

this study.

Thank you for this comment. However, it Is not clear which conclusions the referee refers to. We did mention in Section 3.3.3 (L765-770) "The peak values in the observational and simulated AOD data occur in about the same months, but the simulated maxima are much higher than those of the observations. A reason for this discrepancy may be the large influence of smoke on aerosols in the PRD (Zhang et al., 2010; Liu et al., 2021) which in the CESM model is treated as anthropogenic emissions and thus fixed at the 2010 level (Section 2.3). The data in Fig. 8 show that the simulated AOD in March 2010 was substantially higher than the observed AOD, suggesting that the initial anthropogenic emission estimates were high."

and in Section 4.1 (L956-958): " During these later years, elevated AOD peaks are observed over the PRD in the spring which are attributed to the transport of biomass burning plumes from Indochina during specific weather patterns (Xue et al., 2025)".